# LKRSDH-dependent histone modifications of *insulin-like peptide* sites contribute to age-related circadian rhythm changes

Pengfei Lv [1], Xingzhuo Yang [1] & Juan Du [1] ✉

To understand aging impact on the circadian rhythm, we screened for factors influencing circadian changes during aging. Our findings reveal that *LKRSDH* mutation significantly reduces rhythmicity in aged flies. RNA-seq identifies a significant increase in *insulin-like peptides* (*dilps*) in *LKRSDH* mutants due to the combined effects of H3R17me2 and H3K27me3 on transcription. Genetic evidence suggests that *LKRSDH* regulates age-related circadian rhythm changes through *art4* and *dilps*. ChIP-seq analyzes whole genome changes in H3R17me2 and H3K27me3 histone modifications in young and old flies with *LKRSDH* mutation and controls. The results reveal a correlation between H3R17me2 and H3K27me3, underscoring the role of *LKRSDH* in regulating gene expression and modification levels during aging. Overall, our study demonstrates that *LKRSDH*-dependent histone modifications at *dilps* sites contribute to age-related circadian rhythm changes. This data offers insights and a foundational reference for aging research by unveiling the relationship between LKRSDH and H3R17me2/H3K27me3 histone modifications in aging.

Circadian rhythm is a universal phenomenon in living organisms that helps them better adapt to their environment. Since the discovery of the first gene regulating circadian rhythm in 1972[1], *Drosophila* has become a model system for studying circadian rhythm. In *Drosophila*, the core regulatory loop that controls circadian rhythm consists of CLOCK (CLK) and CYCLE (CYC), which form a heterodimer complex and activate rhythmic transcription of clock-controlled genes[2], as well as PERIOD (PER) and TIMELESS (TIM), which are the main repressors of CLK/CYC activity[3]. Studies have shown that this mechanism is conserved in vertebrates[3].

Age has an impact on circadian rhythm. In *Drosophila*, aged flies exhibit weakened sleep/activity rhythm, as studies have shown[4–6]. Electrophysiological recordings of large ventral lateral neurons in *Drosophila* have demonstrated a significant decrease in input resistance with age. Furthermore, aging in *Drosophila* leads to a reduction in circadian rhythm and the daily plasticity in the architecture of small ventral lateral neurons[7]. In addition, the oscillations of the clock genes *per*, *tim* and their proteins are attenuated in the heads of aged flies compared to young flies[5,8]. Similar changes have been

observed in the clock-associated genes *Pdp1ε, vri*, and *cry*[5,8]. In mammals, it has been also observed that the amplitude of circadian rhythm tends to decrease with age[9,10]. By facilitating SIRT1-dependent deacetylation of the core clock protein PER2, nicotinamide adenine dinucleotide (NAD + ) has the potential to rejuvenate both circadian gene expression and behavior in aged mice[11]. A genome-wide association study has also indicated that the muscle circadian clock is important for longevity[12]. The identification of more factors that regulate changes in the circadian rhythm during the aging process plays a crucial role in unraveling the mechanisms underlying this phenomenon.

Multiple studies have provided evidence for the existence of molecular connections between aging and circadian rhythm. For example, mutations in genes associated with circadian rhythm have been identified to impact the aging process in *Drosophila*[13–15]. In vertebrates, the core circadian system is intricately linked with various aging-associated signaling pathways, notably including mTOR, SIRT1, and AMPK which are nutrient sensors[16]. Gaining a comprehensive understanding of the regulatory mechanisms underlying longevity and

[1]Department of Entomology and MOA Key Lab of Pest Monitoring and Green Management, College of Plant Protection, China Agricultural University, Beijing 100193, China. ✉e-mail: dujuan9981@cau.edu.cn

age-related changes in circadian rhythm will enable us to establish the connection between these two processes.

Chromatin features undergo changes during the aging process. DNA modifications, levels of histone expression, histone methylation and histone acetylation all exhibit alterations throughout aging[17–19]. Genomic instability increases with age, resulting from elevated rates of mutations, transposon instability and transcriptional instability[17]. Epigenetic regulation is also important in governing circadian rhythm[20–23]. Recent research has highlighted that age-related changes in the epigenetic regulation of the clock gene *Per1* contribute to certain facets of cognitive decline[24]. However, the specific regulators responsible for these epigenetic features involved in aging process have yet to be identified.

Histone arginine methylation is regulated by methyltransferases and cofactors. H3R17 is a histone arginine site that can be methylated by coactivator-associated arginine methyltransferase 1 (CARM1)[25]. It has been observed that the stability of CARM1 in the aging heart of mammals is significantly reduced[26], suggesting a potential involvement of histone arginine methylation in the aging process. dLKR/SDH, which is an evolutionarily conserved protein from *Arabidopsis* to humans[27], is a dual-activity enzyme with an N-terminal LKR domain and a C-terminal SDH domain. In *Drosophila*, LKRSDH interacts with histone H3 to inhibit the ecdysone-induced expression of cell death genes by blocking histone H3R17me2, a modification catalyzed by the *Drosophila* arginine methyltransferase ART4 (ortholog of CARM1)[28]. It is still an open question whether histone arginine methylation plays a role in regulating aging.

To investigate age-dependent circadian rhythm changes, we conduct a screening of *Drosophila* mutants, specifically targeting genes involved in the regulation of histone methylation. The mutants are obtained from the Bloomington Stock Center after creating a list of genes related to histone methylation based on a previous publication (Supplementary file 2 in ref. [29]). The results show that the mutation of *LKRSDH* (short for Lysine ketoglutarate reductase/saccharopine dehydrogenase), an inhibitor of histone H3R17me2, leads to significant decrease in rhythmicity among aged flies compared to controls. Combinational analysis of genome-wide differentially expressed genes in *LKRSDH* mutants, as revealed by RNA-seq and ChIP-seq data, indicates that a significant upregulation of *insulin-like peptides*, resulting from the combined effects of H3R17me2 and H3K27me3 on transcription. Further genetic evidence indicates that the regulation of age-related changes in circadian rhythm by *LKRSDH* is mediated by *art4* and *dilps*. A combination of RNA-seq and ChIP-seq is applied to identify the whole genome changes in H3R17me2 and H3K27me3 histone modifications in young and old flies with *LKRSDH* mutations, as well as in controls. The results provides insights into the relationship between *LKRSDH* and histone modifications of H3R17me2 and H3K27me3 during the aging process.

## Results

### *LKRSDH* regulates the circadian rhythm of aged *Drosophila*

As *Drosophila melanogaster* undergoes the aging process, there is a gradual weakening observed in its circadian rhythm[5,6]. To investigate the influence of epigenetic factors on circadian rhythm during aging, we performed a screen for mutants of epigenetic regulators that had a significant impact on circadian rhythm of older flies. Our findings revealed that the rhythmicity of the 40-day-old *LKRSDH*^MB01942 homozygous allele was more severely dampened compared to the 40-day-old *w*^1118 control group (Fig. 1a and Supplementary Fig. 1a–h, *w*^1118 and *LKRSDH*^MB01942 homozygous allele at 40 day). These results strongly indicate the involvement of *LKRSDH* in the regulation of the circadian rhythm in aging flies. Real time RT-PCR analysis confirmed that the expression of *LKRSDH* was significantly down-regulated in the *LKRSDH*^MB01942 homozygous allele (Supplementary Fig. 1i). Western blot analysis also showed a down-regulation of LKRSDH

protein levels in both heterozygous and homozygous alleles (Supplementary Fig. 1j). The antibody used were also validated by a tagged LKRSDH protein (Supplementary Fig. 1k). In comparison to the *w*^1118 control, the aged *LKRSDH*^MB01942 homozygous allele exhibited a more significant decline in power values (Fig. 1b, *w*^1118 and *LKRSDH*^MB01942 homozygous allele at 40 day). Notably, the characteristic morning-evening activity peak pattern was almost entirely absent in the aged *LKRSDH*^MB01942 homozygous allele (Supplementary Fig. 1d, h). In contrast, *LKRSDH*^MB01942 homozygous allele did not have a significant effect on the circadian rhythm parameters (percentage of rhythmicity, tau and power) of 3-day-old *Drosophila* (Fig. 1a, b, *w*^1118 and *LKRSDH*^MB01942 at 3 day). To further validate our findings, we utilized three independent *LKRSDH* RNAis driven by the *LKRSDH*-Gal4 lines, which resulted in similar phenotypes (Fig. 1a, b). These findings provide evidence for an age-dependent impact of *LKRSDH* on the circadian rhythm in flies, with the absence of *LKRSDH* exacerbating the disorder in the circadian rhythm of these flies.

To investigate whether *LKRSDH* regulates the circadian rhythm of aging flies by affecting their activity ability, we conducted an analysis of the activity of both mutant and control groups. We observed a significant decrease in activity counts (number of activities per 30 min) in both the *w*^1118 control and *LKRSDH*^MB01942 homozygous allele during aging (Supplementary Fig. 1l). While the 3-day-old *LKRSDH*^MB01942 homozygous allele exhibited significantly reduced activity counts compared to the *w*^1118 control, this difference was not observed in older flies (Supplementary Fig. 1l). We also examined the locomotor response using the rapid iterative negative geotaxis (RING) assay and found no difference between the 3-day-old *LKRSDH*^MB01942 homozygous allele and the 3-day-old *w*^1118 control (Supplementary Fig. 1m). Consistently, the locomotor response did not show any significant changes in the mutant after aging (Supplementary Fig. 1m), indicating that the *LKRSDH* mutation did not affect the locomotor response.

Interestingly, the circadian rhythm under the 12hrs/12hrs light dark (LD) conditions was significantly affected in *LKRSDH*^MB01942 homozygous allele in both young and aged flies, with a significant change in power value specifically observed in aged flies (Supplementary Fig. 1n, o).

To confirm the regulatory effect of *LKRSDH* on circadian rhythm during aging, we performed rescue experiments using transgenic flies generated through the CRISPR/Cas9 transcriptional activation method to conditionally express *LKRSDH*[30]. *LKRSDH*^TOE.GS01508, referred to as *LKRSDH* OE, specifically activates the *LKRSDH* gene expression in the presence of Gal4[30]. As anticipated, induction of *LKRSDH* expression by *LKRSDH*-Gal4 in 40-day-old flies led to a rescue of both the percentage of rhythmicity and power value of the *LKRSDH*^MB01942 mutant (Fig. 1c, d, *LKRSDH*^MB01942, *UAS-dCas9/LKRSDH-Gal4, LKRSDH* OE in 40d). Genotypes of *LKRSDH*^MB01942, *UAS-dCas9/+* and *LKRSDH-Gal4, LKRSDH* OE were used as controls for *LKRSDH*^MB01942, *UAS-dCas9/LKRSDH-Gal4, LKRSDH* OE. In 3-day-old *Drosophila*, the percentage of rhythmicity and power value of *LKRSDH*^MB01942, *UAS-dCas9/LKRSDH-Gal4, LKRSDH* OE was not significantly changed compared to *LKRSDH*^MB01942, *UAS-dCas9/+*. However, in 40-day-old *Drosophila*, there was a significantly increased in both the percentage of rhythmicity and power value of *LKRSDH*^MB01942, *UAS-dCas9/LKRSDH-Gal4, LKRSDH* OE when compared to *LKRSDH*^MB01942, *UAS-dCas9/+* and *LKRSDH-Gal4, LKRSDH* OE (Fig. 1c, d). In conclusion, supplemented *LKRSDH-Gal4*-driven *LKRSDH* can rescue the phenotypes caused by *LKRSDH*^MB01942 in 40-day-old flies.

### The expression of *LKRSDH* in both neurons and glial cells is necessary for maintaining circadian rhythm in aged flies

To determine the expression pattern of *LKRSDH* in the adult fly brain, we performed florescence in situ hybridization (FISH) experiments. The results indicated that the *LKRSDH* had a ubiquitous expression pattern in the fly brain, some of which was represented by *LKRSDH*-

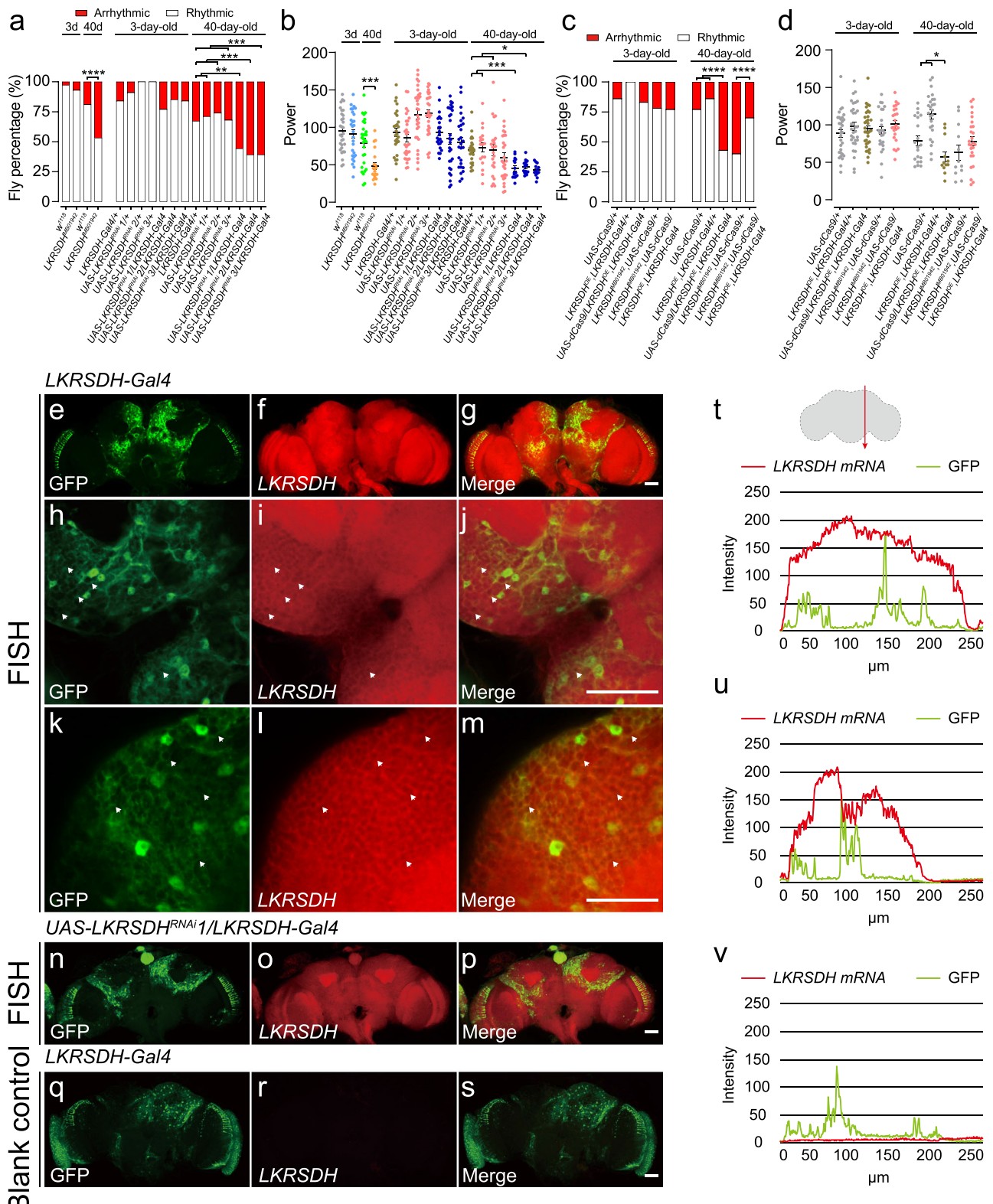

Gal4 (Fig. 1e–m, t). The specificity of the FISH signals was verified by the decrease of signals in *LKRSDH* RNAi driven by *LKRSDH*-Gal4 (Fig. 1n–p, u). The blank control experiment yielded no detectable signals (Fig. 1q–s, v). The data indicated that the *LKRSDH*-Gal4 expressed in a portion of the *LKRSDH* expressing cells. It is common for Gal4 lines of this nature, created through the insertion of a Gal4-expressing cassette into a specific gene's genomic region, only captures a fraction of the gene's overall expression in many cases. This

phenomenon has been previously confirmed in published studies as well (Fig. 2f–h in ref. 31).

We also performed immunofluorescence staining experiments to verify the identity of GFP-expressing cells of the *LKRSDH*[MBO1942] allele, which is an insertional mutation carrying a GFP fluorescent marker[32]. We used anti-Repo and anti-Elav antibodies to label glial cells and neurons, respectively, in the brains of *Drosophila*[33,34]. We found that *LKRSDH* was expressed in a portion of glial cells (Fig. 2a–o, rectangular

**Fig. 1 | *LKRSDH* down regulation or overexpression leads to circadian rhythm defects. a** Percentage of rhythmic and arrhythmic flies (from left to right, bars represent n = 32, 30, 36, 38, 32, 32, 31, 28, 30, 34, 34, 39, 24, 31, 41, 34, 44 and 33 independent samples). *LKRSDH*^MB01942^ are *LKRSDH* homozygous mutant flies. *P*-values from two-sided Fisher's exact test are indicated, where **P < 0.01, ***P < 0.001, and ****P < 0.0001. All the *P*-values are listed in Supplementary Data 3. **b** Circadian power value of flies with rhythm, corresponding to (**a**). Data are presented as mean ± SEM. *P*-values from unpaired two-tailed Student's *t*-test are indicated, *P < 0.05 and ***P < 0.001. If P > 0.05, it will not be presented in the figures. All the *P*-values are listed in Supplementary Data 3. **c** Percentage of rhythmic and arrhythmic flies (from left to right, bars represent n = 36, 31, 41, 32, 31, 26, 29, 28, 30 and 37 independent samples). *LKRSDH*^MB01942^, *UAS-dCas9/+* and *LKRSDH*^MB01942^, *UAS-dCas9/ LKRSDH*^OE^; *LKRSDH-Gal4* are *LKRSDH* heterozygotes mutant flies. *P*-values

from two-sided Fisher's exact test are indicated, where ****P < 0.0001. All the *P*-values are listed in Supplementary Data 3. **d** Circadian power value of flies with rhythm, corresponding to (**c**). Data are presented as mean ± SEM. *P*-values from unpaired two-tailed Student's *t*-test are indicated, *P < 0.05. If P > 0.05, it will not be presented in the figures. All the *P*-values are listed in Supplementary Data 3. **e**–**s** Detection of *LKRSDH* in whole-mount *Drosophila* brains using FISH. Heads of young *Drosophila* were dissected at ZT8. The cells expressing *LKRSDH*-Gal4 were represented by GFP. Scale bar = 10 μm. The arrow highlighted the co localization of *LKRSDH* and GFP. The images presented are from one of the three biological repeats. **t**–**v** The intensity of GFP and FISH signal of *LKRSDH* in samples in panels (**g**), (**p**) and (**s**) were measured using LAS X software. The arrow in the upper portion shows the position of the measurement. Source data are provided as a Source Data file.

box and arrow) and neurons (Fig. 2a–o, round box and arrow). Consistently, the expression of GFP driven by *LKRSDH*-Gal4 also demonstrated widespread *LKRSDH* expression in both neurons and glial cells of the fly brain (Supplementary Fig. 2a–j).

To determine whether *LKRSDH* expressed by neurons or glial cells is required for circadian rhythm regulation during aging, we used *LKRSDH* RNAi to knock down *LKRSDH* in glial cells and neurons. The efficiencies of the RNAis were verified through RT-PCR analysis (Supplementary Fig. 2k–m). The downregulation of *LKRSDH* in glial cells or neurons did not affect the circadian rhythm of young flies (Fig. 2p). Nevertheless, with the progression of aging, the percentage of rhythmicity of flies with *LKRSDH* downregulated in both neurons and glial cells was significantly decreased compared to the controls, which was consistent with the phenotype exhibited by the *LKRSDH*^MB01942^ allele (Fig. 2p, Fig. 1a). Calculation of the power value of the rhythmic flies in these genotype groups did not show any significant changes (Supplementary Fig. 2n). These results indicated that *LKRSDH* expression in both neurons and glial cells was required for *Drosophila* to maintain rhythm during aging.

Overexpression of *LKRSDH* in neurons and glial cells using *nsyb*-Gal4 and *repo*-Gal4, respectively, led to additional functions. In both young and aging flies, overexpression of *LKRSDH* using *nsyb*-Gal4 resulted in a further decrease in the power value and percentage of rhythmicity (Fig. 2q and Supplementary Fig. 2o), while overexpression using *repo*-Gal4 led to a further decrease in the power value (Fig. 2q and Supplementary Fig. 2o). In contrast, overexpression of *LKRSDH* using *LKRSDH*-Gal4 showed no significant difference in the percentage of rhythmicity compared to *LKRSDH*-Gal4 controls in young flies. As flies aged, the percentage of rhythmicity in flies overexpressing *LKRSDH* using *LKRSDH*-Gal4 was found to be significantly lower compared to the *LKRSDH*-Gal4 control group (Fig. 1c, d).

In the rescue experiment, the expression of *LKRSDH* in both neurons and glial cells of the *LKRSDH*^MB01942^ allele led to notably low level of rhythmicity in aged flies, particularly when utilizing *nsyb*-Gal4 (Fig. 2r and Supplementary Fig. 2p). However, young flies were not significantly affected by expressing *LKRSDH* using *nsyb*-Gal4 and *repo*-Gal4 (Fig. 2r and Supplementary Fig. 2p). These findings deviate from the results observed with the *LKRSDH*-*Gal4* expression of *LKRSDH* in the *LKRSDH*^MB01942^ allele, which exhibited a more effective rescue of the mutant phenotype (Fig. 1c, d). These results align with the observation that *LKRSDH* is expressed in both neurons and glial cells (Fig. 2). Consequently, ectopic overexpression of *LKRSDH* in either all neurons or glial cells alone cannot effectively rescue the phenotype caused by the *LKRSDH*^MB01942^ allele.

In conclusion, our findings highlight the indispensable role of *LKRSDH* expression in both neurons and glial cells for the maintenance of circadian rhythm in aged flies. As evidenced by the expression patterns, *LKRSDH* is consistently present in both neurons and glial cells. Consistently, overexpression of *LKRSDH* in either all neurons or glial cells alone results in additional functions but fails to rescue the phenotype caused by the *LKRSDH*^MB01942^ allele.

## Genome-wide RNA-seq revealed the downstream genes of *LKRSDH*

To investigate the downstream target genes of *LKRSDH* involved in regulating the circadian rhythm during aging, we performed genome-wide RNA-seq analysis in both *w*^1118^ control and *LKRSDH*^MB01942^ mutant at the ages of 3 days and 40 days at the time point of ZT8. Our analysis revealed that the majority of differentially expressed genes (DEGs) were upregulated in both young and aged *LKRSDH*^MB01942^ flies (Fig. 3a and Supplementary Data 1). In the *w*^1118^ control group, there were more downregulated DEGs after aging compared to upregulated DEGs (Fig. 3a and Supplementary Data 1). Conversely, in the *LKRSDH*^MB01942^ mutant group, the number of downregulated DEGs after aging exhibited only a marginal decrease compared to the number of upregulated DEGs (Fig. 3a and Supplementary Data 1). These findings provide further support for the role of *LKRSDH* as a transcriptional regulator in ecdysone signaling[28].

Further analysis of overlapping DEGs revealed features in the DEG profiles associated with both the *LKRSDH* mutation and aging process. There were fewer DEGs in the young flies than in the aged flies of *LKRSDH*^MB01942^ allele (Fig. 3a), indicating a heightened sensitivity of aging flies to *LKRSDH* mutation. Comparative analysis showed that 656 DEGs in the aged *LKRSDH*^MB01942^ mutant overlapped with those in the aged *w*^1118^ control, comprising more than half of the total DEGs (Fig. 3a). Notably, only 42 DEGs caused by *LKRSDH*^MB01942^ allele in young flies were shared with those caused by aging in *w*^1118^ control (Fig. 3a). These findings highlight the distinct DEG profiles resulting from aging and *LKRSDH* mutation.

Nevertheless, despite the disparities in the profile of DEGs, KEGG enrichment analysis revealed remarkably consistent pathway enrichment results. This analysis indicated that the *LKRSDH* allele in flies led to alterations in genes associated with the longevity regulation pathway, both in young and old flies (Fig. 3b and Supplementary Data 2). Consistently, aging in the *w*^1118^ control also led to gene changes in the longevity regulation pathway (Fig. 3b and Supplementary Data 2). However, the *LKRSDH* mutation did not lead to significant enrichment of this group of genes, suggesting a potential relationship between *LKRSDH* and aging (Fig. 3b and Supplementary Data 2).

Gene Ontology (GO) analysis of these DEGs revealed common enrichment in pathways associated with the aging process for both the *LKRSDH*^MB01942^ mutant and *w*^1118^ control (Supplementary Fig. 3a and Supplementary Data 2). These common pathways included ion transmembrane transport, ATP metabolism, cellular respiration, etc. (Supplementary Fig. 3a and Supplementary Data 2). The aged *w*^1118^ control exhibited specific enrichment in pathways related to the regulation of membrane potential, cytoplasmic translation, etc. (Supplementary Fig. 3a and Supplementary Data 2). The aged *LKRSDH*^MB01942^ mutant showed specific enrichment in pathways such as carbohydrate derivative catabolic process, chemical homeostasis, neuropeptide signaling pathway, etc. (Supplementary Fig. 3a and Supplementary Data 2).

Among the three gene profiles enriched in the longevity regulating pathway, we identified *dilp3* and *dilp5* on the list (Supplementary

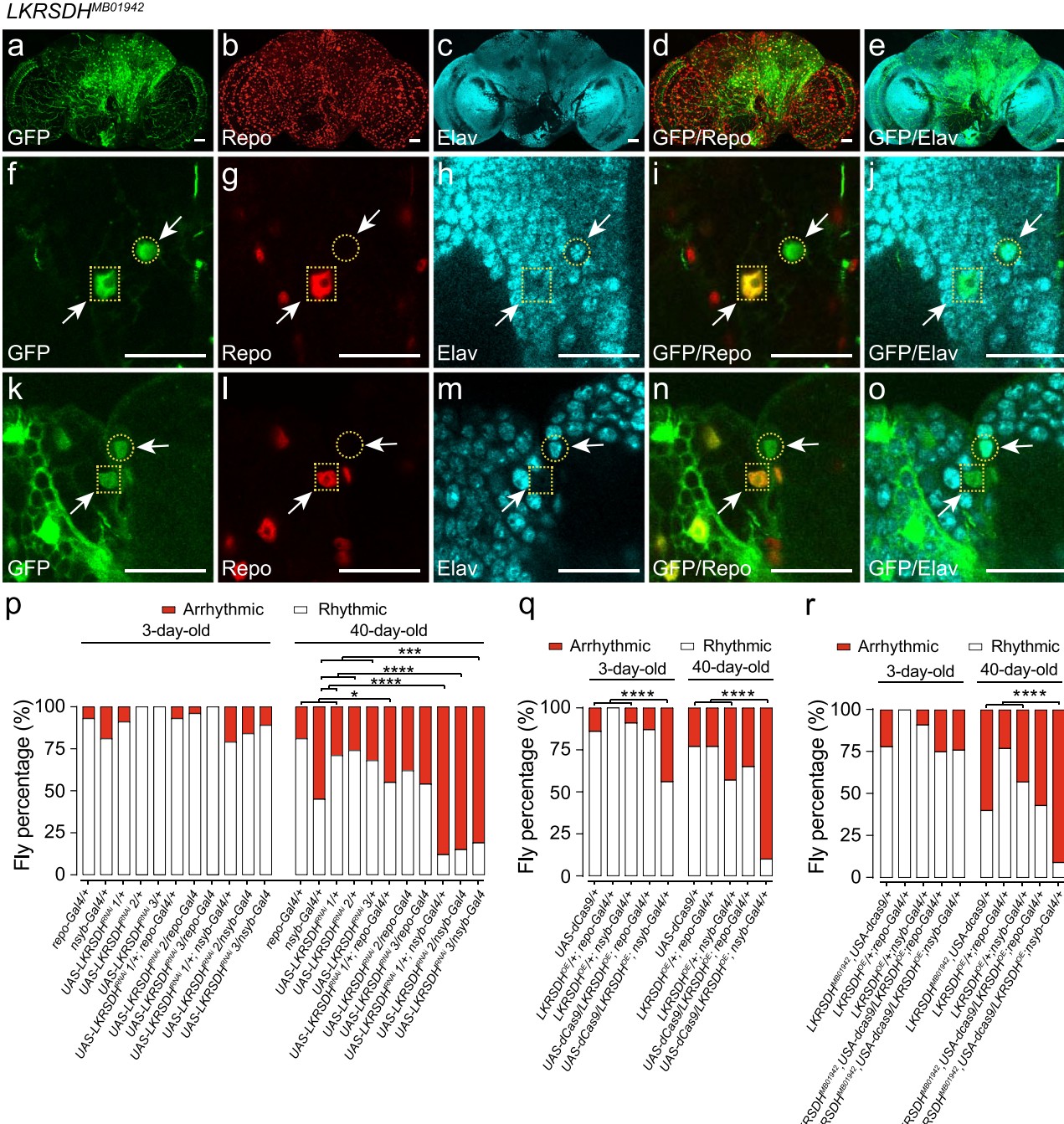

**Fig. 2 | The expression of LKRSDH in both neurons and glial cells is necessary for maintaining the circadian rhythm in aged flies. a–o** LKRSDH represented by GFP in *LKRSDH*$^{MB01942}$ was expressed in the same set of cells with neurons labeled with Elav antibody or glial cells labeled with Repo antibody. Heads of young *LKRSDH*$^{MB01942}$ were dissected at ZT8. The arrow highlights the co-localization of LKRSDH with neurons or glial cells. Cells surrounded by rectangle and circle shapes represent the co-localization of LKRSDH with glial cells and neurons, respectively. Scale bar = 10 μm. The images presented are from one of the three biological repeats. **p** Percentage of rhythmic and arrhythmic flies (from left to right, bars represent *n* = 30, 31, 32, 31, 28, 30, 28, 28, 29, 31, 44, 21, 31, 24, 31, 41, 22, 26, 28, 26, 27 and 27 independent samples). *P*-values from two-sided Fisher's exact test are

indicated, where *\*P* < 0.05, \*\*\**P* < 0.001 and \*\*\*\**P* < 0.0001. All the *P*-values are listed in Supplementary Data 3. **q** Percentage of rhythmic and arrhythmic flies (from left to right, bars represent *n* = 36, 27, 42, 38, 32, 26, 30, 30, 34 and 41 independent samples). *P*-values from two-sided Fisher's exact test are indicated, where \*\*\*\**P* < 0.0001. All the *P*-values are listed in Supplementary Data 3. **r** Percentage of rhythmic and arrhythmic flies (from left to right, bars represent *n* = 32, 27, 42, 36, 37, 30, 30, 30, 42 and 34 independent samples). *LKRSDH*$^{MB01942}$,*UAS-dCas9*/+, *LKRSDH*$^{MB01942}$,*UAS-dCas9/LKRSDH*$^{OE}$;*repo-Gal4*/+ and *LKRSDH*$^{MB01942}$,*UAS-dCas9/ LKRSDH*$^{OE}$;*nsyb-Gal4*/+ are *LKRSDH* heterozygotes mutant flies. *P*-values from two-sided Fisher's exact test are indicated, where \*\*\*\**P* < 0.0001. All the *P*-values are listed in Supplementary Data 3. Source data are provided as a Source Data file.

Data 2). We then examined the expression levels of all the *dilp* family members and found that the *dilp2, dilp3* and *dilp5* exhibited changes in their expression levels under these conditions (Fig. 3c, d and Supplementary Fig. 3b). Consequently, we conducted detailed studies on the relationship between *LKRSDH* and these *dilp* genes.

We also examined the oscillation patterns of these *dilps* and the glucose levels across the 24 hours. The results indicated that while there were significant changes in the expression levels, the oscillating patterns of the expression remained unaltered across different conditions (Supplementary Fig. 3c–e). Moreover, the glucose level was

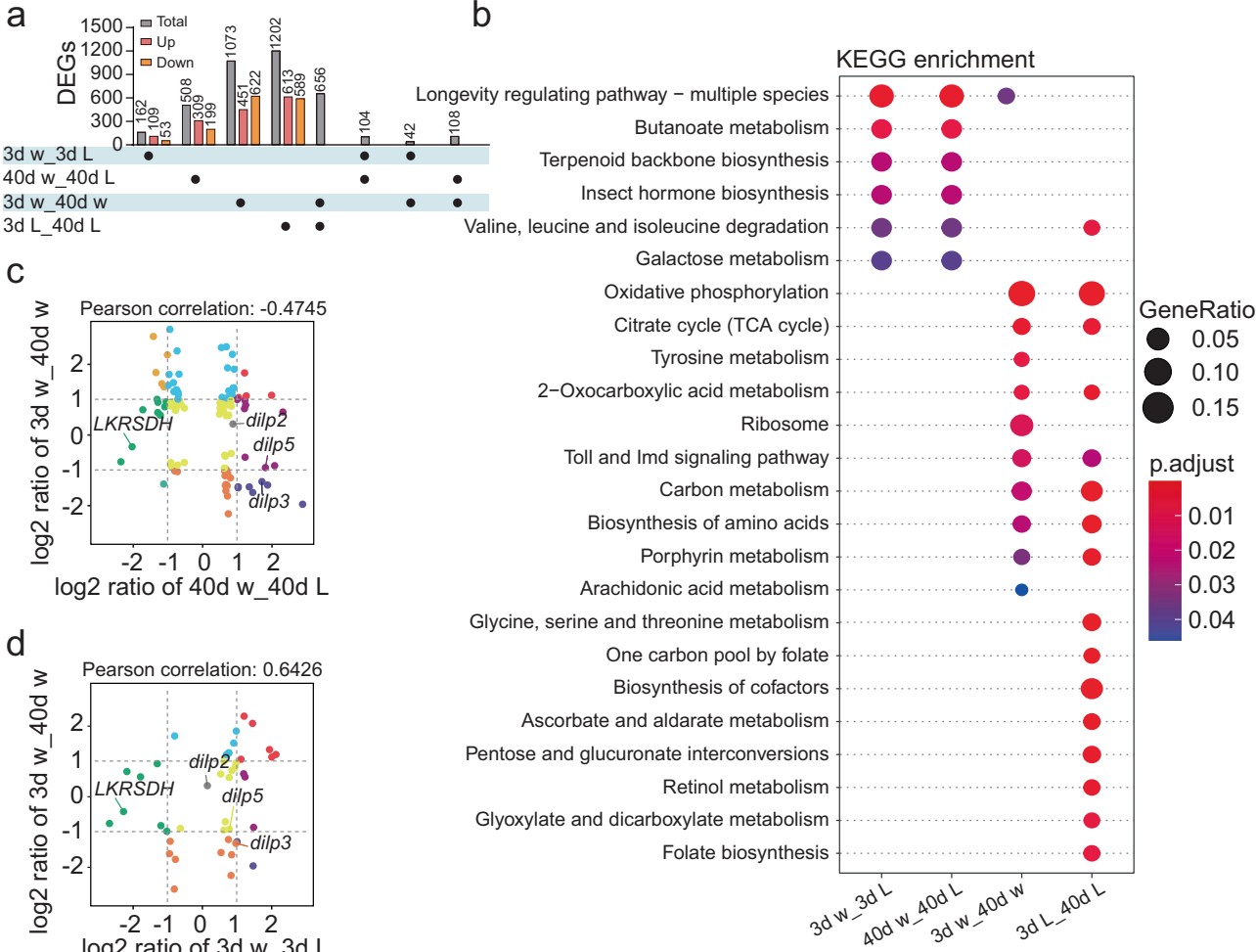

**Fig. 3 | *Dilps* are downstream target genes of *LKRSDH* shown by RNA-seq. a** The number of differentially expressed genes (DEGs) (up-regulated and down-regulated) in different categories. w for *w^III8* and L for *LKRSDH^MBO1942*. Fifty fly heads were collected at ZT8 and used for RNA-seq. **b** KEGG enrichment analysis was performed for genes regulated by *LKRSDH* or altered during aging identified by RNA-seq. KEGG enrichment was measured by adjusted one-tailed *P* values and GeneRatio in this pathway. KEGG terms with p.adjust < 0.05 are shown. **c** The nine-quadrant diagram shows the trend of alteration in shared DEGs in 40d *w^III8* *v.s.* 40d *LKRSDH^MBO1942* and 3d *w^III8* *v.s.* 40d *w^III8*. **d** The nine-quadrant diagram shows the trend of alteration in shared DEGs in 3d *w^III8* *v.s.* 3d *LKRSDH^MBO1942* and 3d *w^III8* *v.s.* 40d *w^III8*. Source data are provided as a Source Data file.

significantly elevated in *LKRSDH* allele compared to the *w^III8* control in young and old flies (Supplementary Fig. 3f). Additionally, within the same genotype, the glucose level was significantly higher in young flies compare to aged flies (Supplementary Fig. 3f). These findings suggest that the *LKRSDH* allele exerts a considerable influence on glucose metabolism in *Drosophila*.

### *Dilp2, dilp3* and *dilp5* contribute to the circadian rhythm and longevity phenotypes caused by *LKRSDH* mutation

As indicated above, *dilp2, dilp3* and *dilp5* were in the list of differentially expressed genes (Fig. 3c, d). Previous evidence has shown that the insulin pathway in *Drosophila* is involved in regulating lifespan[35,36] and circadian rhythm[37]. Therefore, we aimed to investigate whether *dilps* contributed to these phenotypes caused by *LKRSDH* mutation. Through RNA-seq and qRT–PCR, we found that both *dilp3* and *dilp5* were upregulated in the *LKRSDH^MBO1942* allele (Fig. 4a). *dilp2* was upregulated in the *LKRSDH^MBO1942* allele in aged flies, but not in young flies (Fig. 4a). In comparison to the control group, the overexpression of *dilp3* in aged flies decreased the percentage of rhythmicity and the power value, which were consistent with the phenotypes of the *LKRSDH^MBO1942* allele (Fig. 4b and Supplementary Fig. 4a). Compared with the control group, knockdown of *LKRSDH* by *dilp5*-Gal4 resulted in a significant decrease in

the percentage of rhythmicity and the power value of the aged flies (Fig. 4b and Supplementary Fig. 4a). More importantly, although *dilp1-5^KO*/+ did not cause a significant defect in the percentage of rhythmicity, presumably because of heterozygosity, *dilp1-5^KO* rescued the defect in the percentage of rhythmicity caused by the *LKRSDH^MBO1942* allele (Fig. 4c and Supplementary Fig. 4b). In conclusion, these results reveal that *dilps* contribute to the circadian phenotypes caused by *LKRSDH* mutation.

To distinguish the specific roles of *dilp2, dilp3* and *dilp5*, we compared the circadian rhythm phenotypes and the rescue effects of each single mutant. The results indicated that the *dilp2, dilp3* and *dilp5* single mutants exhibited varying degree of defects in the percentage of rhythmicity in both young and old flies (Fig. 4d). The *dilp3 /dilp5* double mutant showed a significant higher percentage of arrhythmicity compared to each single mutant (Fig. 4d). Furthermore, all the *dilp2, dilp3, dilp5* single mutants and *dilp3 /dilp5* double mutant were able to rescue the phenotype caused by the *LKRSDH^MBO1942* allele in aged flies (Fig. 4d). The rescue effects of *dilp2* were better compared to *dilp3* and *dilp5* single mutants, and were comparable to *dilp3 /dilp5* double mutant (Fig. 4d). However, the power value of these genotypes did not show significant changes (Supplementary Fig. 4c). In conclusion, these results collectively reveal that *dilp2, dilp3* and *dilp5* contribute to the circadian phenotypes caused by the *LKRSDH* mutation.

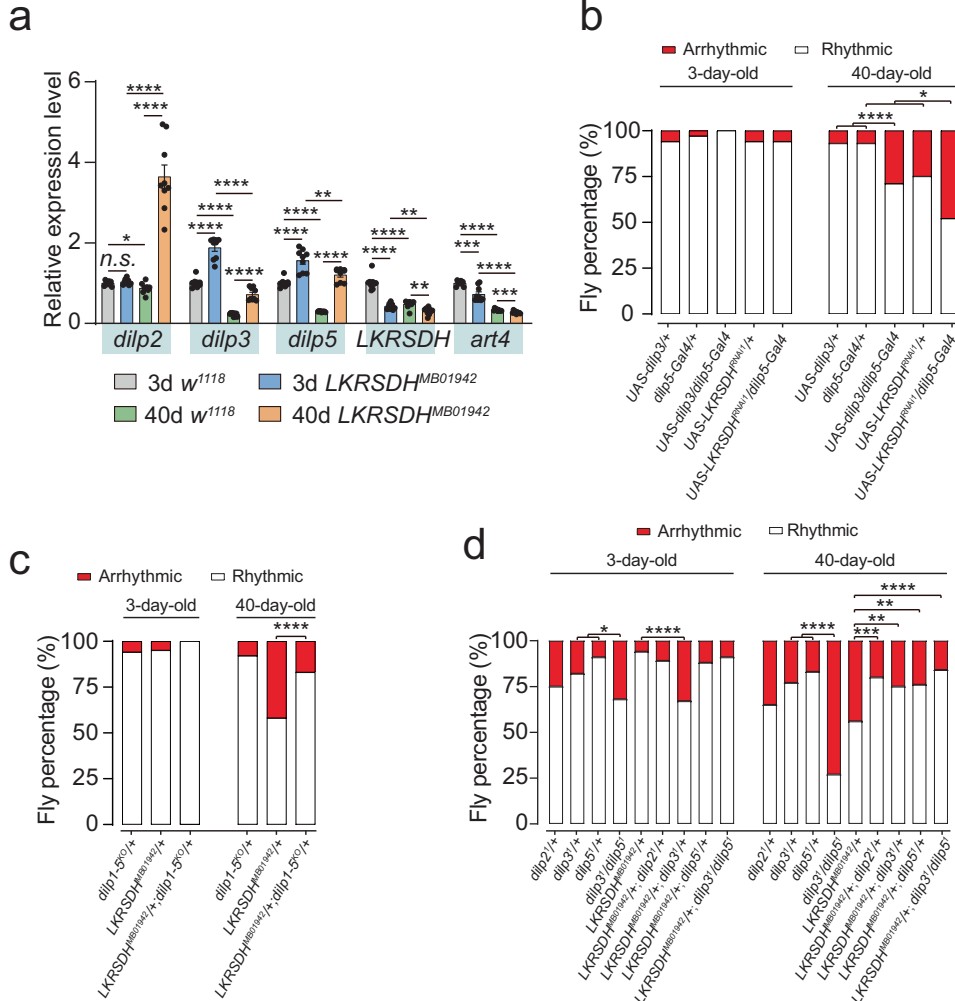

**Fig. 4 | Overexpression or knockout of *dilps* resulted in circadian rhythm defects. a** Quantitative RT-PCR of *dilp2*, *dilp3*, *dilp5*, *LKRSDH* and *art4*. Data are presented as mean ± SEM. *P*-values from unpaired two-tailed Student's *t*-test are indicated, where *n.s.* indicates no significant difference, *$P < 0.05$, **$P < 0.01$, ***$P < 0.001$ and ****$P < 0.0001$. All the *P*-values are listed in Supplementary Data 3. *n* = 3 biologically independent experiments. **b** Percentage of rhythmic and arrhythmic flies (from left to right, bars represent *n* = 32, 32, 32, 31, 31, 30, 27, 24, 28 and 29 independent samples). *P*-values from two-sided Fisher's exact test are indicated, where *$P < 0.05$ and ****$P < 0.0001$. All the *P*-values are listed in Supplementary Data 3. **c** Percentage of rhythmic and arrhythmic flies (from left to right, bars represent *n* = 31, 37, 26, 26, 38 and 28 independent samples). *LKRSDH^MB01942^/+*

and *LKRSDH^MB01942^/+; dilp1-5^KO^/+* are *LKRSDH* heterozygotes mutant flies. *P*-values from two-sided Fisher's exact test are indicated, where ***$P < 0.001$. All the *P*-values are listed in Supplementary Data 3. **d** Percentage of rhythmic and arrhythmic flies (from left to right, bars represent *n* = 28, 27, 32, 37, 32, 27, 46, 33, 58, 26, 31, 30, 33, 27, 30, 28, 29 and 32 independent samples). *LKRSDH^MB01942^/+ , LKRSDH^MB01942^/+; dilp2^1^/+, LKRSDH^MB01942^/+; dilp3^1^/+, LKRSDH^MB01942^/+; dilp5^1^/+, LKRSDH^MB01942^/+; dilp3^1^/dilp5^1^* are *LKRSDH* heterozygotes mutant flies. *P*-values from two-sided Fisher's exact test are indicated, where *$P < 0.05$, **$P < 0.01$, ***$P < 0.001$ and ****$P < 0.0001$. All the *P*-values are listed in Supplementary Data 3. Source data are provided as a Source Data file.

Subsequent experiments provided further evidence regarding the genetic relationship mentioned above and its impact on the longevity phenotype. Detection of the longevity phenotype of the *LKRSDH* allele indicated that it shortened the lifespan (Supplementary Fig. 4d). The expression of *LKRSDH* rescued this shortened lifespan phenotype (Supplementary Fig. 4e). More importantly, *dilp1-5* knockout also rescued the longevity phenotype caused by the *LKRSDH* allele (Supplementary Fig. 4f). Overexpressing *dilp3* caused a decreased lifespan (Supplementary Fig. 4g). These results suggest that *dilp3* and *dilp5* are downstream of *LKRSDH* and play a crucial role in regulating both circadian rhythm and longevity.

### *LKRSDH* regulates *dilp2*, *dilp3* and *dilp5* through H3R17me2 and H3K27me3

In addition to its enzymatic activity, *LKRSDH* also functions as a transcriptional corepressor in ecdysone signaling. It inhibits the methylation of histone H3 on R17 by CARMER (ART4 in *Drosophila*), resulting

in the downregulation of gene expression through chromatin remodeling[28,38,39]. Building upon this understanding, we conducted experiments to investigate the potential involvement of *art4* in this regulatory process. Overexpression of *art4* by *LKRSDH*-Gal4 resulted in defects in the percentage of rhythmicity in both young and old flies (Fig. 5a). The power value was also affected in young flies overexpressing *art4* (Fig. 5b). More importantly, the knockdown of *art4* by *LKRSDH*-Gal4 led to significant changes in the percentage of rhythmicity of old flies compared to other Gal4s (Fig. 5a). Consistently, knocking down *art4* by *dilp5*-Gal4 resulted in a suppression of the phenotype caused by *LKRSDH* allele in aged flies (Fig. 5c). The power value showed more decrease in young flies, while the down regulation of the power value by *LKRSDH* allele in aged flies was not suppressed by the *art4* RNAi in aged flies (Fig. 5d compared with Fig. 1b). These results indicate that the *art4* activity is up regulated in *LKRSDH* allele. Consistently, the *dilp* expression levels could be affected by up or down regulation of *art4* (Fig. 5e, f). Therefore, we hypothesize that

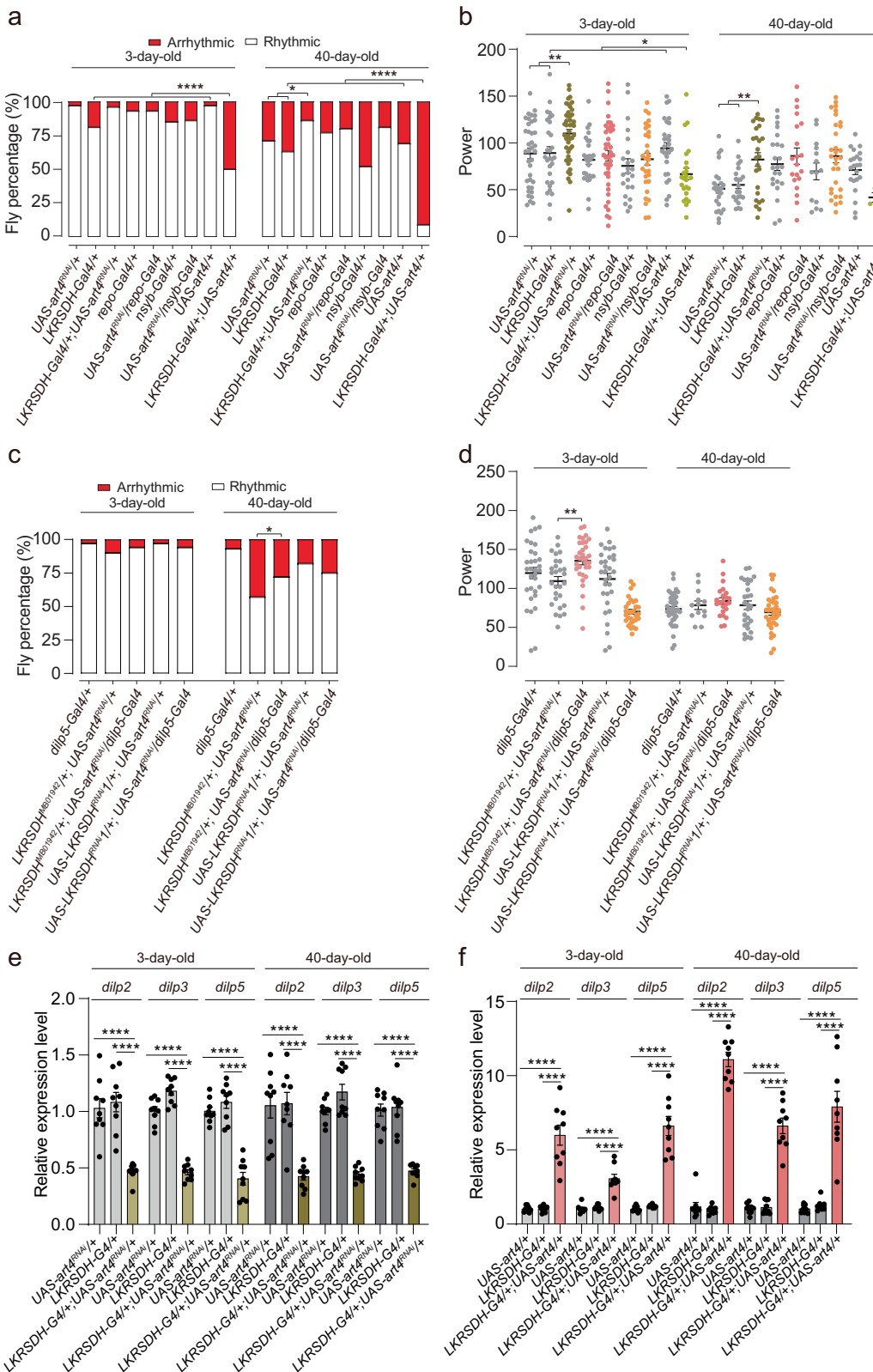

*LKRSDH* regulates *dilp2*, *dilp3* and *dilp5* through *art4*. To investigate this hypothesis, we conducted an examination of various histone modifications, including H3R17me2, among others.

Subsequently, we investigated the impact of the *LKRSDH* mutation on the global level of H3R17me2 and found that its levels were elevated in young *LKRSDH^MB01942* mutant flies (Fig. 6a, c). However, H3R17me2 was significantly decreased in aged *w^1118* and *LKRSDH^MB01942*

mutant flies (Fig. 6a, c). Subsequent results demonstrated that the decline in H3R17me2 levels observed in aged flies was attributed to a decrease in the expression level of *art4* mRNA (Fig. 4a). We speculated that the decrease in H3R17me2 contributes to the down-regulation of most genes during aging, including *dilps* (Fig. 4a). Furthermore, in the young *LKRSDH^MB01942* flies, the decrease in *art4* transcription (Fig. 4a) may be a result of feedback regulation in

**Fig. 5 | LKRSDH regulates circadian rhythm through art4. a** Percentage of rhythmic and arrhythmic flies (from left to right, bars represent $n = 38, 34, 46, 30, 44, 27, 35, 29, 48, 35, 35, 29, 30, 25, 27, 37, 32$ and $34$ independent samples). *P*-values from two-sided Fisher's exact test are indicated, where \**P* < 0.05 and \*\*\*\**P* < 0.0001. All the *P*-values are listed in Supplementary Data 3. **b** Circadian power value of flies with rhythm, corresponding to (**a**). Data are presented as mean ± SEM. *P*-values from unpaired two-tailed Student's *t*-test are indicated, where \**P* < 0.05 and \*\**P* < 0.01. If P > 0.05, it will not be presented in the figures. All the *P*-values are listed in Supplementary Data 3. **c** Percentage of rhythmic and arrhythmic flies (from left to right, bars represent $n = 33, 31, 35, 32, 32, 40, 23, 32, 34$ and $48$ independent samples). *LKRSDH^{MB01942}/ + ; UAS-art4^{RNAi}/+ and LKRSDH^{MB01942}/ + ; UAS-art4^{RNAi}/dilp5-Gal4*

are *LKRSDH* heterozygotes mutant flies. *P*-values from two-sided Fisher's exact test are indicated, where \**P* < 0.05. All the *P*-values are listed in Supplementary Data 3. **d** Circadian power value of flies with rhythm, corresponding to (**c**). Data are presented as mean ± SEM. *P*-values from unpaired two-tailed Student's *t*-test are indicated, where \*\*\*\**P* < 0.0001. If P > 0.05, it will not be presented in the figures. All the *P*-values are listed in Supplementary Data 3. **e, f** Quantitative RT-PCR of *dilp2, dilp3* and *dilp5*. Data are presented as mean ± SEM. *P*-values from unpaired two-tailed Student's *t*-test are indicated, where \*\*\*\**P* < 0.0001. All the *P*-values are listed in Supplementary Data 3. $n = 3$ biologically independent experiments. Source data are provided as a Source Data file.

response to *LKRSDH* depletion and the more efficient methylation of histone H3R17 by *art4*.

To further investigate whether *LKRSDH* regulates *dilps* through histone modifications, we examined the co-expression of *LKRSDH* and *dilps* in the same set of cells. The immunostaining results showed that *dilp5*-Gal4-driven *Redstinger* was expressed in the same set of cells as *LKRSDH*, labeled by the GFP trap in *LKRSDH^{MB01942}* (Fig. 6d–i). Furthermore, we analyzed H3R17me2 in gene body and promoter regions of *dilps* (Fig. 6j–m). We found that in the young *LKRSDH^{MB01942}* mutant, H3R17me2 was significantly increased at the *dilps* promoter region, which correlated with elevated *dilps* expression (Fig. 4a). Surprisingly, we also found a significant increase in H3R17me2 levels at the *dilps* gene body and promoter in the aged *w^{1118}* control (Fig. 6k–m). Moreover, H3R17me2 was significantly downregulated in old flies carrying the *LKRSDH^{MB01942}* mutation (Fig. 6k–m). Neither of these findings fully explained the changes in *dilps* expression during aging (Fig. 4a). Hence, we hypothesized that other histone modifications could be involved in *dilps* regulation. Previous results have shown that CARM1 plays a role in modulating the antagonism between EZH2 and BAF155 to facilitate the H3K27me3 deposition by EZ in cancer cells[40]. In the modENCODE database, we identified binding sites of H3K27me3 in the *dilps* gene body and promoter regions (Supplementary Fig. 5a). Therefore, we conducted tests for H3K27me3.

The detection of H3K27me3 at the global level revealed its dependency on *LKRSDH*. H3K27me3 increased significantly during the aging process of *w^{1118}* control (Fig. 6b, c). *LKRSDH* mutation led to a significant increase in H3K27me3 in young flies (Fig. 6b, c). In old flies, there was no significant change in the global level of H3K27me3 in the *LKRSDH^{MB01942}* mutant (Fig. 6b, c). In the *LKRSDH^{MB01942}* mutant, the aging process resulted in significant increase in H3K27me3 (Fig. 6b, c).

We also examined H3K27me3 on the *dilps* gene body and promoter. We found that the *LKRSDH^{MB01942}* allele did not affect H3K27me3 in young flies. However, as the flies aged, the H3K27me3 modification at the *dilps* gene body and promoter regions decreased compared with the *w^{1118}* control (Fig. 6j–m). Moreover, H3K27me3 was significantly increased in the *dilps* gene body and promoter region in the aged *w^{1118}* control (Fig. 6j–m). These results shed light on the mechanism underlying changes in *dilps* expression levels. Specifically, in young flies, *LKRSDH^{MB01942}* allele caused an increase in H3R17me2 at the *dilps* locus, without affecting H3K27me3. As a result, this promoted the expression of *dilps*. As the flies aged, the *LKRSDH^{MB01942}* allele induced a significant decrease in both H3R17me2 (activating mark) and H3K27me3 (repressive mark) at the *dilps* locus compared to the aged *w^{1118}* control, resulting in increased *dilps* expression (Supplementary Fig. 5b–d). Consistently, in aged flies, although H3R17me2 of the *dilps* locus in the *LKRSDH^{MB01942}* mutant decreased compared with the *w^{1118}* control, it appeared that H3K27me3 at the *dilps* locus decreased relatively more, potentially contributing to the increased expression of *dilps* (Supplementary Fig. 5b–d). Interestingly, overexpressing *LKRSDH* in neurons or glia resulted in down regulation of H3R17me2 (Supplementary Fig. 5e–g). Although there were no notable changes in H3K27me3 observed in young flies, a significant increase was detected in aged flies (Supplementary Fig. 5h–j).

Interestingly, the changes in the levels of H3R17me2 and H3K27me3 at specific gene loci were not always consistent with the global levels. In aged flies, when mRNA of *art4* was greatly downregulated, the downregulation of *LKRSDH* did not result in global changes in H3R17me2 and H3K27me3 levels (Figs. 4a, 6c). However, at the *dilp3* and *dilp5* gene loci, the downregulation of *LKRSDH* led to decreased H3R17me2 and H3K27me3 levels in aged flies (Fig. 6k–m), suggesting the existence of specific regulatory mechanisms for the two histone modifications at particular gene loci. Subsequently, we asked to what extent H3R17me2 associated with H3K27me3 and the effect of aging and *LKRSDH* on this association.

The alterations of H3R17me2 genome widely and at the gene locus supported the regulatory relationship between *LKRSDH* and *art4*. The downregulation of *art4* resulted in a decrease in H3R17me2 levels (Supplementary Fig. 6a–d). While, the upregulation of *art4* resulted in an increase in H3R17me2 levels (Supplementary Fig. 6e–h). Consistently, the downregulation of *art4* further decreased H3R17me3 in the presence of the *LKRSDH^{MB01942}* allele (Supplementary Fig. 6i–k), indicating the *art4* was downstream of *LKRSDH*. However, the alterations of H3K27me3 genome widely and at the gene locus indicated the mechanisms in addition to *art4* may be involved in mediating the effects of *LKRSDH* on H3K27me3. Downregulation of *art4* in neurons resulted in an increase in H3K27me3 (Supplementary Fig. 7a–d). While, the upregulation of *art4* resulted in a decrease in the H3K27me3 (Supplementary Fig. 7e–g). If the *art4* were the only factor mediating the effects of *LKRSDH* on H3K27me3, *LKRSDH* mutant which results in the upregulation of *art4* activity should have led to a decrease in H3K27me3. However, global H3K27me3 were either upregulated or not significantly changed by the *LKRSDH* allele, which arguing the presence of other mechanisms (refer to the Discussion session). This data also suggested that the H3K27me3 levels changes caused by *LKRSDH* mutation on the *dilp* gene sites in aged flies were dependent on *art4*. The downregulation of *art4* led to further up regulation of H3K27me3 in the presence of the *LKRSDH^{MB01942}* allele (Supplementary Fig. 7h–k). Interestingly, the downregulation of *art4* resulted in the decrease of *LKRSDH*, suggesting the existence of a feedback mechanism at transcriptional level (Supplementary Fig. 8).

### H3R17me2 is positively correlated with H3K27me3, and both are widely repressed by the *LKRSDH* genome

To clarify the relationship between H3R17me2 and H3K27me3, we conducted chromatin immunoprecipitation followed by sequencing (ChIP-seq) experiments on H3R17me2 and H3K27me3 in 3-day-old *w^{1118}* control, 3-day-old *LKRSDH^{MB01942}* mutant, 40-day-old *w^{1118}* control and 40-day-old *LKRSDH^{MB01942}* mutant flies at ZT8. Through these experiments, we obtained maps representing the distribution of H3R17me2 and H3K27me3 along chromosomes. We found that H3R17me2 exhibits more extensive occupancy across the genome (Fig. 7a, b). Consistently, H3R17me2 and H3K27me3 predominantly bind to the promoter region (Supplementary Fig. 9a). Further analysis indicated that the majority of binding sites for H3R17me2 and H3K27me3 are located within a 1 kb radius around transcriptional start sites (TSSs) (Supplementary Fig. 9b). In addition, the peak count frequency signal

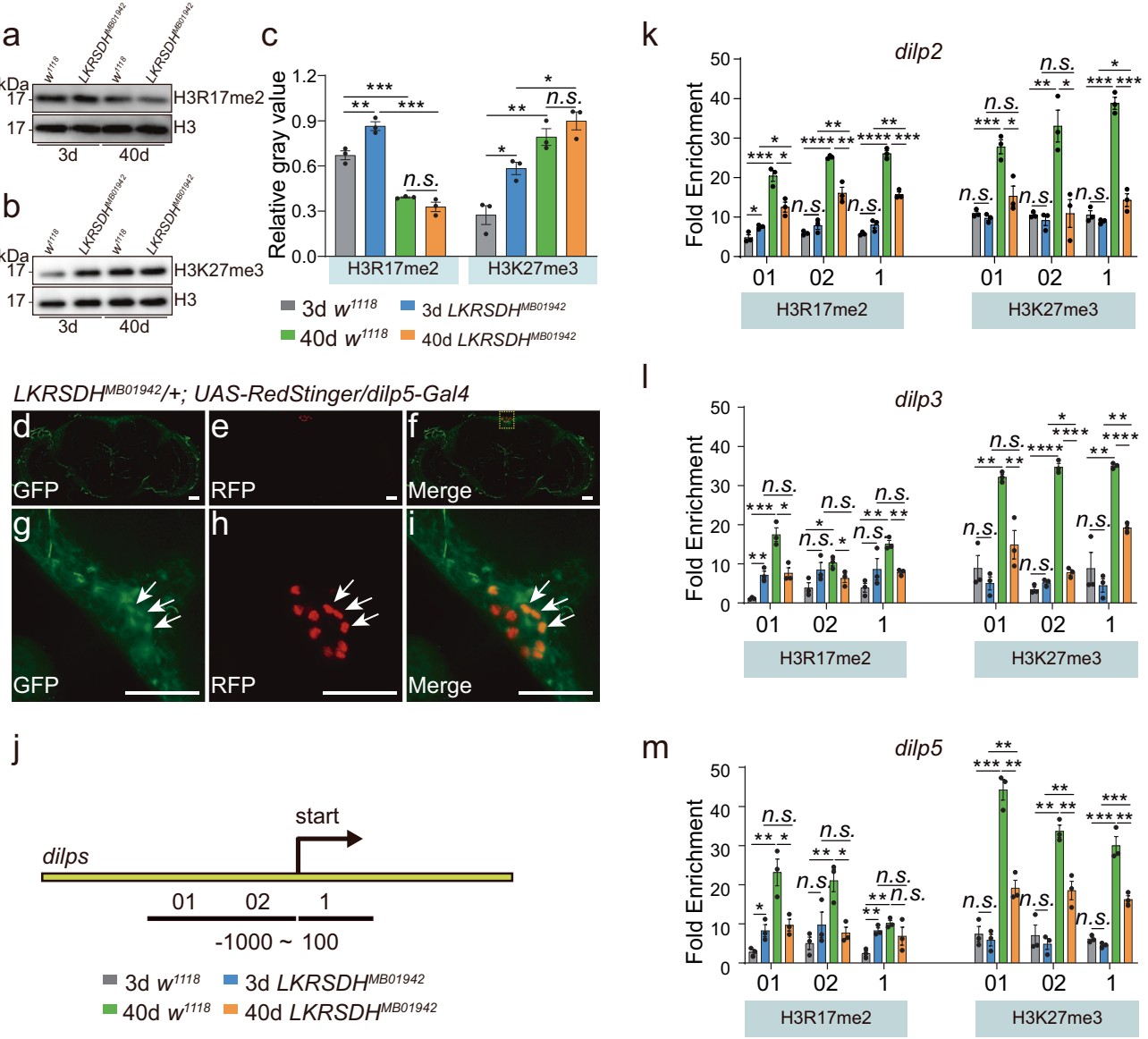

**Fig. 6 | LKRSDH regulates *dilps* through H3R17me2 and H3K27me3. a, b** *LKRSDH* regulates the dynamics of histone modifications H3R17me2 and H3K27me3. Western blot analysis was conducted on H3R17me2 and H3K27me3, with Histone H3 serving as the loading control. The images presented are from one of the three biological repeats. **c** The relative gray values of Western blot in (**a, b**) were measured using Image J. Data are presented as mean ± SEM. *P*-values from unpaired two-tailed Student's *t*-test are indicated, where *n.s.* represents no significant difference, * represents *p* < 0.05, ***p* < 0.01 and ****p* < 0.001. All the *P*-values are listed in Supplementary Data 3. *n* = 3 biologically independent experiments. **d–i** LKRSDH represented by GFP in *LKRSDH^MB01942^* was expressed in the same set of cells with *dlip5*-Gal4 driven *Redstinger*. Scale bar = 10 μm. Heads of *LKRSDH^MB01942^/+ ;UAS-RedStinger/dilp5*-Gal4 were dissected at ZT8. The square area in (**d–f**) is enlarged in (**g–i**). The arrow indicates that LKRSDH is expressed in the cells where *dlip5* is

expressed. The images presented are from one of the three biological repeats. **j–m** ChIP experiments on *dilp2*, *dilp3* and *dilp5* locus. *n* = 3 biologically independent experiments. **j** Schematic diagram of the *dilps* gene region and the location of primers used for ChIP analysis. Primers were within 1000 bp upstream of the *dilps* transcription start (−1000 - 0 bp) and 100 bp downstream of the *dilps* transcription start (0 - 100 bp), respectively. The bent arrow indicates the transcription initiation site. **k–m** The fold enrichment of two histone modifications, H3R17me2 and H3K27me3, on *dilps* promoter and gene body. Data are presented as mean ± SEM. *P*-values from unpaired two-tailed Student's *t*-test are indicated, where *n.s.* represents no significant difference, *represents *p* < 0.05, ***p* < 0.01, ****p* < 0.001 and *****p* < 0.0001. All the *P*-values are listed in Supplementary Data 3. Source data are provided as a Source Data file.

profiles of H3R17me2 and H3K27me3 showed increased average enrichment scores in the regions upstream and downstream of the TSS (Supplementary Fig. 9c). *LKRSDH^MB01942^* mutation or aging altered the distribution and occupancy of H3R17me2 and H3K27me3 (Fig. 7a–d).

The data obtained from ChIP-seq and RNA-seq analyses at the *dilp2*, *dilp3* and *dilp5* gene loci confirmed the findings presented in Fig. 4. In 3-day-old flies, *LKRSDH^MB01942^* allele promoted the enrichment of H3R17me2 at the gene bodies and promoters of *dilp3* and *dilp5*, thereby resulting in increased transcription of *dilp3* and *dilp5* mRNA (Figs. 4a, 6l, m, 7e–g). Aging promoted the enrichment of H3R17me2

and H3K27me3 at the gene bodies and promoters of *dilp2*, *dilp3* and *dilp5*, leading to a decline in *dilp2*, *dilp3* and *dilp5* mRNA transcription (Figs. 4a, 6k–m, 7e–g). This decrease may be attributed to higher levels of H3K27me3 binding. In 40-day-old flies, *LKRSDH^MB01942^* allele led to decreased H3R17me2 and H3K27me3 levels at the gene bodies and promoters of *dilp2*, *dilp3* and *dilp5*, which collectively resulted in elevated *dilp2*, *dilp3* and *dilp5* mRNA transcription (Figs. 4a, 6k–m, 7e–g). While, the levels of H3R17me2 and H3K27me3 at the *dilp1* and *dilp4* locus were not significantly affected under the same conditions (Supplementary Fig. 9d, e).

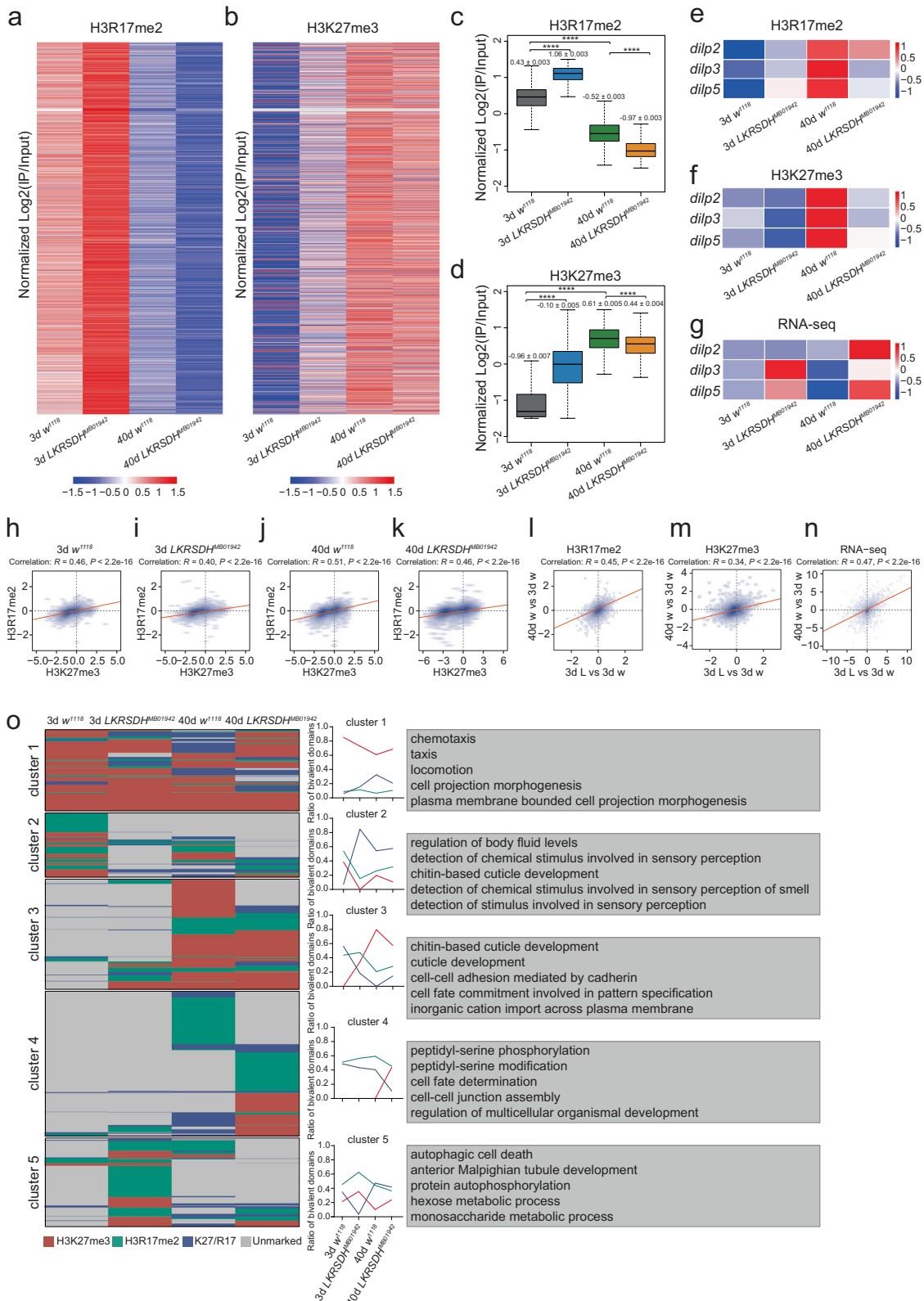

To investigate the relationship between H3R17me2 and H3K27me3 occupancy across the genome, we conducted a correlation analysis based on the peak data generated by H3R17me2 and H3K27me3[41]. The results showed a weak positive correlation at a genome-wide scale (Fig. 7h–k). Surprisingly, the correlation between H3R17me2 and H3K27me3 appeared to increase with aging, suggesting a potential concordance in their partial functions during the aging process (Fig. 7h–k). We also examined the levels of correlation between H3K27me3 and other types of histone modifications using data published previously (see *Methods* for details). The level of correlation between H3R17me2 and H3K27me3 was comparable to the correlation between H3K36me3 and H3K27me3 (Supplementary Fig. 9g–v). As expected, the level of correlation between H3K9me3 and H3K27me3 was relatively high, while the levels of correlation

**Fig. 7 | Analysis of ChIP-seq data revealed that H3R17me2 is positively correlated with H3K27me3. a, b** Heatmaps showing the Log$_2$FC of H3R17me2 and H3K27me3 across the gene body, based on normalized ChIP-seq read coverage. Twenty-five fly heads were collected at ZT8 and used for ChIP-seq. **c, d** Boxplot showing the Log$_2$FC value of H3R17me2 and H3K27me3 across the gene body, respectively, Corresponding to (**a, b**). n = 13967 genes. The middle line within the box plots corresponds to the median; the lower and upper hinges correspond to the first and third quartiles; the upper whisker extends from the hinge to the largest value no further than 1.5 times the interquartile range; and the lower whisker extends from the hinge to the smallest value at most 1.5 times the interquartile range. *P*-values from unpaired two-tailed Student's *t*-test are indicated, where ****$P < 0.0001$. **e, f** Heatmap showing H3R17me2 and H3K27me3 enrichment at *dilp2, dilp3* and *dilp5* locus based on normalized ChIP-seq read coverage. **g** Heatmap showing read counts by RNA-seq at *dilp2, dilp3* and *dilp5* locus. **h–k** Genome-wide correlation plots of H3R17me2 and H3K27me3 occupancies at 3-day-old $w^{1118}$, 3-day-old *LKRSDH$^{MB01942}$*, 40-day-old $w^{1118}$

and 40-day-old *LKRSDH$^{MB01942}$*. The correlation coefficient (*R*) is indicated. **l, m** Genome-wide correlation plots of H3R17me2 and H3K27me3 occupancies at 3-day-old *LKRSDH$^{MB01942}$* / 3-day-old $w^{1118}$ and 3-day-old $w^{1118}$ / 40-day-old $w^{1118}$, respectively. **n** Genome-wide correlation plots of RNA-seq signal at 3-day-old *LKRSDH$^{MB01942}$* / 3-day-old $w^{1118}$ and 3-day-old $w^{1118}$ / 40-day-old $w^{1118}$. **o** Heatmaps showing the dynamics of H3K27me3, H3R17me2 and H3K27me3/H3R17me2 co-modified genes in 3-day-old $w^{1118}$, 3-day-old *LKRSDH$^{MB01942}$*, 40-day-old $w^{1118}$ and 40-day-old *LKRSDH$^{MB01942}$* (left panel). The ratios of H3K27me3, H3R17me2 and H3K27me3/H3R17me2 co-modified genes are also shown (middle panel). The red line indicates the fraction of domains that is marked only by H3K27me3, the green line represents the fraction of domains that is marked only by H3R17me2, and the blue line indicates the fraction of H3K27me3/H3R17me2 co-modified genes. The top five GO enrichment of different gene clusters is shown on the right panel. Source data are provided as a Source Data file.

between H3K4me3 or H3K27ac and H3K27me3 were very low (Supplementary Fig. 9g–v).

Correlation analysis was conducted to examine the changes in H3R17me2 or H3K27me3 occupancy peaks during aging, comparing them with those caused by *LKRSDH*. The analysis revealed the involvement of *LKRSDH* in regulating the aging process. The weak correlation ($R = 0.45$) provides support for the role of *LKRSDH* in regulating H3R17me2 during the aging process (Fig. 7l). Similarly, the weak correlation ($R = 0.34$) also elucidates the important role of *LKRSDH* in regulating H3K27me3 during aging (Fig. 7m). Consistently, the correlation analysis of the DEGs indicated a weak correlation in the gene alterations resulting from *LKRSDH$^{MB01942}$* allele or aging (Fig. 7n), further confirming the involvement of *LKRSDH* in the regulation of aging.

Cluster analysis was performed on the data obtained from ChIP-seq and RNA-seq to identify genes with similar levels of modifications and expression under different conditions. It was observed that when genes displayed low levels of H3K27me3, their expression levels were correlated with H3R17me2 levels. For example, Cluster 1 included genes involved in ncRNA metabolic process, Cluster 5 consisted mainly of genes related to mRNA metabolic process, and cluster 6 was primarily associated with cytoplasmic translation. These clusters demonstrated a correlation between low-to-high expression levels and high-to-low levels of H3R17me2 (Supplementary Fig. 9f). However, when genes exhibited high H3K27me3 levels, the expression of this gene class was not favored, regardless of the H3R17me2 levels. This was observed in Clusters 2, 3, and 4 (Supplementary Fig. 9f). Cluster 2 consisted mainly of genes involved in mitotic cell cycle process, Cluster 3 mainly encompassed genes associated with sensory perception of chemical stimulus and Cluster 4 included genes related to positive regulation of transcription by RNA polymerase II. Moreover, based on the expression level and binding of H3K27me3 and H3R17me2, it was found that H3K27me3 played a dominant role in gene expression regulation (Supplementary Fig. 9f), where the gene expression level depended on H3K27me3. Collectively, these results highlight the relevance of genome-wide H3R17me2 and H3K27me3 levels and their relationship with gene regulation.

Cluster analysis of data obtained from ChIP-seq in different samples revealed the biological features of genes with similar trends of changes in modifications under different conditions (Fig. 7o). Cluster 1 to cluster 5 represented five types of trends of changes (Fig. 7o). For example, in cluster 1, genes enriched in chemotaxis and locomotion had relatively low levels of H3R17me2. The H3K27me3 was down-regulated by *LKRSDH* mutation in 3-day-old flies, while the opposite was observed in 40-day-old flies. The trend of changes in H3R17me2/H3K27me3 co-modified genes was opposite to that of genes with only H3K27me3 modifications. In cluster 2, which included genes enriched in body fluid regulators, chemical stimulus detector etc., H3R17me2 and H3K27me3 were down regulated after *LKRSDH* mutation in 3-day-old flies. H3K27me3 was also down regulated after *LKRSDH* mutation in

40-day-old flies, while the H3R17me2 was upregulated. H3R17me2/H3K27me3 co-modified genes were up regulated after *LKRSDH* mutation in both 3-day-old and 40-day-old flies. These data indicated that the changes in H3R17me2 and H3K27me3 in *LKRSDH* mutation and during aging were site specific.

In conclusion, we found that in fly heads, gene expression and H3R17me2 and H3K27me3 levels changed during the aging process. Genetic evidence also demonstrated that *dilp3* and *dilp5* were major mediators of *LKRSDH* mutant phenotypes during aging. More importantly, our investigation unveiled a correlation between gene expression levels, H3R17me2 levels and H3K27me3 levels in the alterations induced by *LKRSDH* mutation and by aging, highlighting the role of *LKRSDH* in mediating the aging process, which was proven by mutant phenotypes. These conclusions are summarized in a diagram (Fig. 8). Based on our data, we propose that the age dependent phenotype of *LKRSDH* mutant is attributed to the exceptionally high levels of *dilps*, including *dilp2, dilp3, dilp5*, which are presented in aged *LKRSDH* mutant. Data presented in this study collectively establish a mechanism that indicates *LKRSDH* regulates the expression of downstream genes by affecting *art4* activity dependent H3R17me2 and H3K27me3.

## Discussion

In this study, we conduct a genetic screen to identify epigenetic factors that regulate the circadian rhythm of aged flies. We discover that *LKRSDH*-dependent histone modifications of *insulin-like peptide* genes play a role in age-related circadian rhythm changes. This study reports the regulation of H3R17me2 in both aging and circadian rhythm. Through RNA-seq analysis, we identify the differentially expressed genes profile of *LKRSDH* mutant in young and aged flies, thereby identifying *LKRSDH* downstream targets. More importantly, by examining the H3K17me2 and H3K27me3 profiles of the *LKRSDH* mutant and control groups in both young and aged flies in the whole genome, we unveil the correlation between H3R17me2 and H3K27me3 during the aging process. The global features of H3R17me2 and H3K27me3 in the *LKRSDH* mutant and control groups highlight the role of *LKRSDH* in the aging process. Therefore, our study provides valuable insights into the relationship between *LKRSDH* and H3R17me2/ H3K27me3 histone modifications during the aging process.

These findings provide valuable resources for future studies on aging and the nervous system. While aging is a physiological process that affects various tissues in the body, each tissue exhibits common and unique features during aging[42,43]. Although our genome-wide assays are performed in the *Drosophila* head, we observe an impact on lifespan in multiple mutants and double mutants (Supplementary Fig. 4d–g). Indeed, it remains an open question whether H3R17me2 and *LKRSDH* are involved in the aging process of other tissues.

Our results are consistent with previous studies investigating the impact of *insulin* on aging. Data in this study reveal that *LKRSDH*

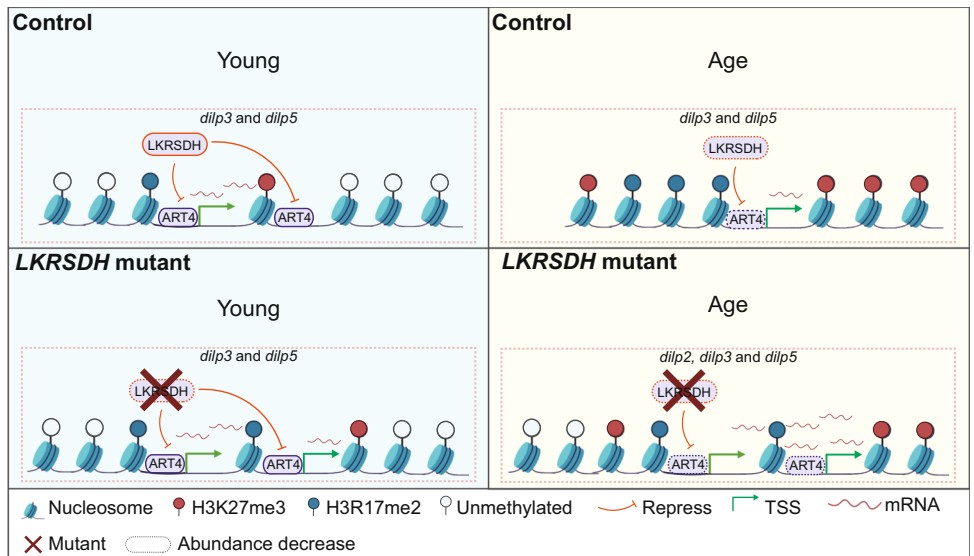

**Fig. 8 | Model for LKRSDH function during aging process.** On *dilp* gene locus, *art4* is recruited to the promoters of *dilp2*, *dilp3* and *dilp5*. *LKRSDH* inhibits the ability of *art4* to methylate histone H3 on R17 and K27. In young flies, proper levels of these genes maintain the expression levels of *dilp* genes. In the absence of *LKRSDH*, the H3R17me2 level increases, leading to elevated expression of *dilp3* and *dilp5*. As flies age, the abundance of *LKRSDH* and *art4* significantly decreased, and

H3K27me3 heavily occupied the promoter of *dilp2*, *dilp3* and *dilp5*, leading to a decrease in the expression of *dilp2*, *dilp3* and *dilp5;* whereas when *LKRSDH* is removed in aged flies, the occupation of H3R17me2 and H3K27me3 were significantly reduced compared to the control, thus promoted the expression of *dilp2*, *dilp3* and *dilp5*.

mutation resulted in increased *dilps* and reduced lifespan. Previous research has demonstrated that decreased insulin pathway activity results in a longer lifespan[44–47]. A study in pancreatic cells has indicated that H3R17me2 is involved in the regulation of insulin[48]. Therefore, this study reveals the upstream regulation of *dilps*, which may have conservation and influence in vertebrates.

In terms of the mechanism underlying *LKRSDH*'s regulation of H3K27me3, our findings strongly imply that this regulation is contingent upon the involvement of *art4*. Evidence supporting this includes the observation that the presence or absence of *art4* changes the trend of H3K27me3 at the *dilps* locus in the background of the *LKRSDH* mutant (Supplementary Fig. 7), suggesting *art4* as downstream of *LKRSDH*. In addition, it was found that the regulation of JH by LKRSDH was independent of its enzymatic activity[28]. However, as mentioned in the results section of our study, it is important to acknowledge that other mechanisms may also contribute to this mechanism. For instance, UTX, an alpha-ketoglutarate-dependent H3K27me3 demethylase enzyme, may be influenced by *LKRSDH*, thus affecting H3K27me3 levels. Further investigation could involve manipulating the enzyme-related domain to explore the involvement of *LKRSDH* in modulating UTX.

The neurons responsible for *LKRSDH* regulation on circadian rhythm may include neurons other than *dilp* expressing neurons. Our data indicates that the *LKRSDH*-Gal4 co-localized with *tim*-Gal4 expressing neurons (Supplementary Fig. 10a–f). In addition, knocking down of *LKRSDH* by *tim*-Gal4 driven *LKRSDH* RNAi recapitulates the *LKRSDH* mutant phenotype (Supplementary Fig. 10g, h). These results collectively indicate that the clock neurons are also responsible for *LKRSDH* function.

This study suggests features of changes of histone modifications during aging process. Our study reveals alterations in *LKRSDH* expression and genome-wide H3K27me3 and H3K17Rme2 levels in the *Drosophila* head during the aging process. *LKRSDH* expression is found to decrease during aging. H3K27me3 levels, in accordance with previous reports[49,50], increases with aging. H3R17me2 levels decrease during aging. The *LKRSDH* mutation leads to heightened levels of H3R17me2 and H3K27me3, as well as a shortened lifespan.

These findings indicate that merely reverting the H3R17me2 level alone is insufficient for inhibiting aging. Second, analysis also reveals a strong correlation between features resulting from the mutation of *LKRSDH* and aging process (Fig. 7l–n). The *LKRSDH* mutation leads to an increase in H3K27me3, consistent with changes during aging, and it also cause an increase in H3R17me2, which is inconsistent with aging-related changes. This result suggests that the effects of H3K27me3 on gene expression changes during aging may be more significant than those of H3R17me2. Third, our results indicate that changes in H3R17me2 and H3K27me3 are correlated with aging. Previous studies have shown that histone modifications, such as H4K16 acetylation, H3K4 trimethylation, and H3K9 or H3K27 trimethylation, serve as hallmarks of aging. These modifications may result in changes in gene transcription, disruption of cellular balance, and a decline in metabolic function associated with aging. These findings suggest that H3R17me2 may have similar effects on the aging process as other histone modifications and may also serve as a hallmark of aging. Our study further reveals that H3R17me2 specifically correlates with aging. The regulation of aging-related H3R17me2 changes is influenced by *LKRSDH*. Lifespan is found to be affected by *LKRSDH* mutation. Previous reports have indicated that *art4*, which is responsible for H3R17me2, is linked to the aging process. These pieces of evidence suggest that the mechanism underlying changes in H3R17me2 and H3K27me3 during aging is linked to their writers or modifiers, which may be influenced by alterations in metabolic or signaling pathways outside of the nucleus that occur during the aging process.

The level of H3R17me2 detected in this study is significantly lower than that of H3K27me3. This may be caused by two possible reasons. First, the antibodies used to detect H3R17me2 and H3K27me3 may have different affinities. Second, H3R17me2 may be more specifically distributed within cells compared to H3K27me3. However, further evidence is required to verify these possibilities.

Analysis also reveals a correlation between H3R17me2 and H3K27me3 (Fig. 7h–k). This correlation could be enhanced by *LKRSDH*, as *LKRSDH* mutation promoted both modifications (Fig. 6c). On the other hand, *art4* may inhibit this correlation since it promotes

H3R17me2 modification while inhibiting H3K27me3 modification (Supplementary Fig. 7a). This could explain why the correlation between H3R17me2 and H3K27me3 was downregulated in the *LKRSDH* mutant (compare Fig. 7i, k with 7h, j). Moreover, in aged flies where the expression of *art4* was significantly downregulated (Fig. 4a), we observe an enhancement in the correlation between H3R17me2 and H3K27me3 (compare Fig. 7j, k with 7h, i). It is possible that *art4* plays a more predominant role compared to *LKRSDH*, as the expression of *LKRSDH* is also downregulated during aging (Fig. 4a). Other mechanisms may also contribute to this crosstalk between H3R17me2 and H3K27me3, making it an intriguing area for further investigation.

This study identifies specific genomic loci enriched with H3R17me2 and H3K27me3 during aging. A recent study has suggested that chromatin accessibility increases with aging[18], and the loss of H3K9me3, a constitutive heterochromatin mark, also occurs during aging[18,51]. It is possible that there is a shift in histone modification from H3K9me3, which is a constitutive heterochromatin mark, to H3K27me3, which is a facultative heterochromatin modification during aging. It would be fascinating to examine the correlation between these H3K9me3 genomic loci and H3R17me2/ H3K27me3 in the same tissue during aging.

## Methods

### Fly stocks
$w^{1118}$ (NO.5905), *LKRSDH*$^{MB01942}$ (NO.23382), *LKRSDH-Gal4* (NO.92724), *LKRSDH*$^{OE}$ (NO.79712), *UAS-LKRSDH*$^{RNAi}$3 (NO.35470), *nsyb-Gal4* (NO.51941), *UAS-RedStinger* (NO.8547), UAS-mCD8: GFP (NO.5137) and *dilp1-5*$^{KO}$ (NO.30890) were obtained from the Bloomington stock center. *Repo-Gal4* was a gift from Dr. Yi Rao's laboratory (Peking University, China). *UAS-dilp3* and *dilp5-Gal4* were gifts from Dr. Li Yan's laboratory. *UAS-LKRSDH*$^{RNAi}$1 (NO.THO2254) and *UAS-dCas9-VPR* (NO.THN0701) were from the TsingHua Fly Center. *UAS-LKRSDH*$^{RNAi}$2 (NO.7144R-1) was from National Institute of Genetics (NIG). *UAS-art4*$^{RNAi}$ (NO.V15645) was from the Vienna *Drosophila* Resource Center (VDRC). *dilp2*$^1$ (NO.30881), *dilp3*$^1$ (NO.30882) and *dilp5*$^1$ (NO.30884) were gifts from Dr. Liwei Zhang's laboratory (China Agricultural University, China). *UAS-art4* was generated by germline transformation. Fly stocks were maintained on standard food in an incubator at 25 °C and 60% humidity with a 12 h light/12 h dark cycle. All mutants and transgenes were backcrossed by a $w^{1118}$ for seven generations to clean up the genetic background. Flies for aging were collected and transferred to fresh food every 3 days. All experiments in this study were conducted using male flies.

### *Drosophila* Activity Monitor-based method for circadian rhythm measurement
Circadian rhythm of individual male flies was measured using the *Drosophila* Activity Monitoring (DAM) System (Trikinetics). Male flies (aged 3 and 40 days old) were loaded individually into glass tubes with a length of 65 mm and an inner diameter of 5 mm. The tubes contained standard cornmeal fly food at one end and were sealed with a cotton stopper at the other end. The flies were entrained to a 12 h light/12 h dark cycle for 3 days and then released to constant darkness for at least six days to measure their rhythmicity.

Chi-squared periodogram analysis determined whether a fly was significantly rhythmic ($\alpha = 0.05$)[52]. Data analysis is done on a Macintosh computer running the FaasX (Fly activity analysis suite) software (obtained from the website https://trikinetics.com). Individual flies with power ($\geq 10$) and "width" value of 1.5 or more (representing the peak number within 30 min increments above the 95% confidence line of the periodogram) were considered rhythmic[53]. Activity data was performed using MATLAB (MathWorks, Natick, MA) for each fly were binned every 30 min for the circadian rhythm analyses.

### Cell culture and transient transfection
S2 cells were obtained from ATCC (ATCC, CRL-1963). S2 cells were grown in Serum-Free Insect Cell Culture Medium (HyClone, SH30278.02) at 25 °C. Transient transfection was performed using X-tremeGENE HP DNA Transfection Reagent (Roche, 06366236001), according to the manufacturer's protocol.

### Plasmid constructions
To obtain the *LKRSDH*, PCR amplification was performed using the cDNA from fly heads and specific primers, namely NotI-*LKRSDH* and HindIII-*LKRSDH*. The resulting *LKRSDH* fragment was then inserted into the digested plasmid pAC-4v5, which had been cut with NotI and HindIII. As a result of this molecular manipulation, the final construct pAC-*LKRSDH*−4v5 was generated. All the primers used are listed in Supplementary Table 1.

### *Drosophila* Lifespan assays
Mated males were collected and maintained at a density of 20 flies per vial. The flies were kept at a temperature of 25 °C with 60% humidity and a 12 hr light/12 hr dark cycle. Every other day, the flies were transferred to new vials and their survival rate was scored. Statistical analysis and survival curve plotting were performed using Prism 8.0 (Graphpad). Each experiment was conducted with three biological replicates, and a total of 300 flies were used for lifespan assays.

### Rapid Iterative Negative Geotaxis (RING) Assay
The RING assay was conducted following a similar method as described by ref. 54, with minor modifications[55]. Ten flies were collected under brief CO2 anesthesia for 1–2 min and allowed to recover overnight at a temperature of 25 °C with 65% relative humidity before the assay. To assess RING, the flies were transferred into a tube with a graduated line positioned 5 cm from the bottom. The tube was gently tapped vertically to cause the flies to ascend the wall. The number of flies crossing the graduated line within 10 s was recorded and the percentage was calculated. Three consecutive trials were conducted with a 60-second interval between each trial. It is important to note that, to ensure fairness, the genotypes of the flies were concealed during the experiment and uninformed individuals carried out the RING assay following the experimental protocol. Each experiment was performed with six biological replicates.

### RNA fluorescence in situ hybridization (FISH)
FISH labeling followed the protocol of Long et al.[56]. with the following modifications. After acetic acid treatment and 0.5% PBT washing, the brains were incubated with pre-hybridization buffer (RIBOBIO, C10910) at 37 °C for 2 h. Then the brains were incubated with FISH probes (Labeled with Cy3 and synthesized by RIBOBIO biological company, dilute to 1:25 with hybridization buffer) in 37 °C for 48 h according to the manufacturer's instructions. After the series of wash steps described in the FISH protocol, the tissues were mounted with DPX and imaged using confocal microscopy (Leica SP8) with LAX software. GFP with a 488-nm laser and 498–543 nm detection, Cy3 with a 561-nm laser and 569–595 nm detection. The laser intensity is uniformly 10% and FISH intensity is measured using LAS X.

### Immunofluorescence experiments of *Drosophila* brain
The flies, which were three days old, were first anesthetized with $CO_2$. Subsequently, their brains were dissected in PBST (1 × PBS with 0.03% Triton X-100; coolaber, CT11451) and fixed with 4% paraformaldehyde at room temperature (RT) for 50 min. After fixation, the brains were washed in PBST three times at RT for 15 min each time. Following this, the brains were blocked in 10% Normal Goat Serum (NGS, diluted with 1 × PBS with 2% Triton) overnight at 4 °C. The brains were then incubated with primary antibodies for 36 h on a rotator at 4 °C. Subsequently, the brains were washed again three times for 15 min in PBST at

RT before being incubated with secondary antibodies for 12 h on a rotator at 4 °C. Once again, the brains were washed three times at RT with PBST and mount on a slide using an anti-fading Mounting medium (Solarbio, S2110). Finally, the brains were imaged using confocal microscopy (Leica SP8) with LAX software.

The primary antibodies used in this experiment were anti-Elav rat (DSHB, 7E8A10, 1:200) and anti-Repo mouse (Developmental Studies Hybridoma Bank, 8D12; 1:200). The secondary antibodies employed were Alexa Fluor™ 568 (Thermo Fisher Scientific, A11004; 1:200) and Alexa Fluor™ 647 (Thermo Fisher Scientific, A21247; 1:200). Both primary and secondary antibodies were diluted in dilution buffer consisting of 1.25% PBST and 1% NGS.

### Western Blots

Fly heads were collected at ZT8 and used for Western Blot analysis. Total protein was extracted from approximately 30 fly heads using RIPA buffer (150 mM NaCl, 1.0% NP-40, 0.5% sodium deoxycholate, 0.1% SDS, 50 mM Tris-HCl, pH 8.0). A protease inhibitor cocktail (Roche, 04693132001) was added to the buffer according to the manufacturer's instructions. The protein concentration was determined using a BCA protein assay kit (Solarbio, PC0020). 20 μg of each sample was separated on 15% SDS–polyacrylamide mini-gels, and transferred onto 0.22 μm PVDF membranes (Millipore, ISEQ85R). The corresponding primary antibodies were diluted 1:1000 and incubated with the membranes overnight at 4 ˚C, while the HRP-labeled secondary antibodies were diluted 1:1000 and incubated for 4 h at room temperature. The signals were detected using ECL (ABclonal, RM00021P) and visualized using the Amersham ImageQuant 800 (GE Healthcare, Sweden). After detecting the signal of histone modification, we used stripping buffer (0.1 L: 20 mL SDS 10%; 12.5 mL Tris HCl, pH 6.8, 0.5 M; 67.5 mL distilled water and add 0.8 mL ß-mercaptoethanol under the fume hood) to remove the primary and secondary antibodies from the western blot membrane. The purified membrane was subsequently incubated with anti-Histone H3 and secondary antibody, resulting in the detection of the H3 signal. To generate the anti-LKRSDH signals, the PVDF membrane was incubated with LKRSDH antibody. The primary and secondary antibodies were then removed using a stripping buffer, after which the membrane was incubated with v5 antibody to generate specific anti-v5 signals. The primary antibodies used in this study were anti-Histone H3 (asymmetric di methyl R17) (Abcam, ab8284; 1:500), anti-Histone H3 (tri methyl K27) (Abcam, ab6002; 1:1000), anti-Histone H3 (EASYBIO, BE3015; 1:1000), anti-LKRSDH antibody (Abconal, A24544; 1:50), anti-β-tubulin (Abconal, AC008; 1:1000) and anti-v5 antibody (Abconal, AE017; 1:1000). The secondary antibodies used in this study were HRP Goat Anti-Rabbit IgG (H + L) (ABconal, AS014; 1:1000) and HRP Goat Anti-Mouse IgG (H + L) (ABconal, AS003; 1:1000). The signal intensity was quantified using ImageJ software (NIH). The relative normalization of histone modifications was performed with Histone H3 as controls. Three biological replicates were conducted for each experiment.

### Quantitative RT−PCR (qRT-PCR)

Total RNA was extracted from approximately 30 fly heads using TRIzol Reagent (vazyme, R401-01). The cDNAs were synthesized by reverse transcription using HiScript III All-in-one RT SuperMix (Vazyme, R333), with 1 μg total RNA used as a template for each sample. The cDNA preparation was then subjected to real time quantitative PCR (Applied Biosystem, Step One Real-Time PCR system) according to the protocol of ChamQ SYBR qPCR Master Mix (High ROX Premixed) (Vazyme, Q341). Data analyses were performed using Prism 8.0 (Graphpad). All the primers used are listed in Supplementary Table 1.

### Chromatin immunoprecipitation (ChIP) assays followed by quantitative PCR

Male fly heads of 3-day-old $w^{1118}$, 3-day-old $LKRSDH^{MB01942}$, 40-day-old $w^{1118}$ and 40-day-old $LKRSDH^{MB01942}$ were collected at ZT8 for ChIP-qPCR

analysis. Briefly, 25 fly heads were gently triturated in 450 μl 1 × PBS, and then 6.02 μl 37% formaldehyde was added for cross-linking. The mixture was incubated at RT for 10 min. After de-crosslinking with 1 × glutamate and washing the pellet three times with 1 × PBS, 250 μl SDS lysis buffer (containing protease inhibitor cocktail) was added and homogenized. The solution was then centrifuged, and the supernatant was collected. Chromatin was sonicated on ice for 2.5 min (settings 10 s on, 30 s off, high power), resulting in sheared chromatin with an average length of 0.1–0.5 kb. The lysate was immunoprecipitated overnight at 4 °C with 40 μl Protein A/G Magnetic beads (Thermo Fisher Scientific, 88803) and 2 μg of antibody. Unbound chromatin was discarded, and the beads were washed with Low Salt Wash buffer, High Salt Wash buffer, LiCl Wash buffer and TE Wash buffer. The beads were then resuspended in freshly prepared ChIP elution buffer. DNA was eluted by adding 5 M NaCl and incubating for 4 h in a 65 °C water bath. The obtained DNA was purified and used for qPCR. For this experiment, anti-IgG, anti-Histone H3 (asymmetric di methyl R17) (abcam, ab8284) and anti-Histone H3 (tri methyl K27) (abcam, ab6002) were used. All ChIP analyses were independently repeated three times as biological replicates. Data analyses were performed using Prism 8.0 (Graphpad). All the primers used are listed in Supplementary Table 1.

### RNA-seq, ChIP seq and data analysis

Fly heads of 3-day-old $w^{1118}$, 3-day-old $LKRSDH^{MB01942}$, 40-day-old $w^{1118}$ and 40-day-old $LKRSDH^{MB01942}$ were collected at ZT8 for RNA-seq analysis. The Illumina library preparation and sequencing steps were conducted at the sequencing platform of Biomics Company in Beijing, China (http://www.biomics.com.cn/). Raw sequencing data underwent processing and quality control using Trim-galore (https://github.com/FelixKrueger/TrimGalore) and FastQC (http://www.bioinformatics.babraham.ac.uk/projects/fastqc/). The RNA-seq reads were aligned to the *Drosophila melanogaster* (dm6) reference genome (https://hgdownload.soe.ucsc.edu/goldenPath/dm6/bigZips/) using HISAT2[57]. HTSeq-count[58] was used to count RNA-seq reads, and differentially expressed genes (DEGs) were identified using the R package DESeq2[59] with FDR value < 0.05 and | log$_2$ (fold change) | > 0.5 as criteria. Kyoto Encyclopedia of Genes and Genomes (KEGG) and Gene Ontology (GO) analyses were conducted using the R package Cluster Profiler[60] and org.Dm.eg.db dictionary. Three biological replicates were performed for the RNA-seq experiment.

For ChIP-seq analysis, 25 fly heads of 3-day-old $w^{1118}$, 3-day-old $LKRSDH^{MB01942}$, 40-day-old $w^{1118}$, and 40-day-old $LKRSDH^{MB01942}$ were collected at ZT8. In this experiment, anti-Histone H3 (asymmetric di methyl R17) (abcam, ab8284) and anti-Histone H3 (tri methyl K27) (abcam, ab6002) antibodies were used. The Illumina library preparation and sequencing were performed at the Novogene's (https://cn.novogene.com/) sequencing platform. Adaptor sequence trimming, mapping to the *Drosophila melanogaster* (dm6) reference genome using Bowtie 2[61] and PCR duplicate removal using Picard Tools (http://broadinstitute.github.io/picard/) were carried out. MACS2[62] was used to call broad peaks, and the R package ChIPseeker[63] was employed for peak annotation and analysis. The deepTools bamCompare function[64], multiBigWigSummary function[64] and R package pheatmap were used to generate ChIP-seq signal heatmaps. The RNA-seq signal heatmaps represented the log$_2$-ratio of fold change over the gene body. deepTools multiBigWigSummary function[64] was used to compute average RPKM-normalized read count in peaks, and the Pearson correlation between different ChIP-seq targets was calculated by ggplot2 in Rstudio. H3K27me3/H3R17me2 co-modified genes were those genes with both H3K27me3 and H3R17me2 covered genes. To identify co-modified genes, we derived the H3K27me3 and H3R17me2 covered matrix using the same cut-off during the identification of dynamic histone modification domains (greater than the max H3K27me3 and H3R17me2 signal level of non-marked domains)[65]. We used *k*-means

clustering to cluster different co-modified genes, with k setting to 5 for H3K27me3/H3R17me2 co-modified genes analysis. Heatmap and cluster analysis in Fig. 7o and Supplementary Fig. 9f were generated by R package Complex Heatmap and ClusterGVis, respectively[66,67]. Two biological replicates were conducted for the ChIP-seq experiment.

For Fig. 7h–n, the Pearson correlation coefficient $R$ serves as a measure of the degree of correlation between variables, with values ranging from −1 to 1. A positive value indicates a positive correlation, while a negative value indicates a negative correlation. $R$ values between 0 and 0.25 or between 0 and −0.25 are commonly interpreted as indicating the absence of correlation. $R$ values between 0.25 and 0.50 or between −0.25 and −0.50 suggest a poor correlation between variables. $R$ values ranging from 0.50 to 0.75 or −0.50 to −0.75 indicate a moderate to good correlation, while $R$ values between 0.75 and 1 or between −0.75 and −1 signify a very good to excellent correlation between the variables[41].

### Glucose measurement
Six male flies were collected at ZT8 and homogenized in 250 μL of PBS with 0.2% Triton-X. The homogenates were then incubated at 70 °C for 5 min to inactivate endogenous enzymes. After centrifugation at 13,500 × g for 15 min at 4 °C, 150 μL of supernatant was collected for subsequent measurement. The PGO Enzyme Preparation Solution (Sigma, P7119) was prepared following the manufacturer's instructions. The PGO Enzyme Reaction Solution was prepared by adding water, glucose standard solution (Sigma, G6918), and the glucose-containing sample according to the manufacturer's instructions. All tubes were then incubated at 37 °C for 30 min, and the absorbance (A) of the standards and tests was measured at 450 nm using the Blank as the reference. The glucose concentration of the sample was determined using the following formula:

$$\text{Sample Glucose Concentration (mg/mL)} =$$

$$\frac{\text{Absorbance (Test)} \times \text{Dilution of sample} \times 1\,\text{mg/mL}}{\text{Absorbance(Standard)}}$$

### Statistical analysis
Statistics analysis of rhythmic datasets was performed in Graphpad Prism v.8.0 using Fisher's exact test or an unpaired two-tailed Student's $t$-test. The Log-rank (Mantel-ox) test was used for survival rate analysis. For comparisons of two groups unpaired two-tailed Student's $t$-test was used. The error bars in the related types of figures represent the standard error of mean (s.e.m.). The precise test method used for the different figures is also specified in the corresponding figure legends. $P$-value < 0.05 was considered significant. All the $P$-values are listed in Supplementary Data 3.

### Reporting summary
Further information on research design is available in the Nature Portfolio Reporting Summary linked to this article.

### Data availability
The RNA-seq data generated in this study have been deposited in the National Center for Biotechnology Information (NCBI) database under accession code PRJNA965887. The ChIP-seq data generated in this study have been deposited in the NCBI's Gene Expression Omnibus (GEO) database under accession code GSE235532. Published gene expression datasets used in this study can be find from GEO database under accession code GSE153901, GSE37032, GSE59769, GSE94922[68–71]. All other data needed to reproduce the results presented here are contained within the manuscript, figures, supplementary information and Zenodo database and are accessible to the public through the accession code: 10785489. Source data are provided with this paper.

### Code availability
All the custom computer codes and programs required to generate the data of this work are publicly available in the Zenodo repository and accessible through the [https://zenodo.org/records/10785489][72].

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

## Acknowledgements

This work was supported by National Natural Science Foundation of China (Grant number 32122017 and 32070492) to J.D.

## Author contributions

P.L. designed and performed the experiments, analyzed the data and wrote the manuscript. X.Y. did bioinformatic analysis. J.D. designed experiments, analyzed data and wrote and edited the manuscript.

## Competing interests

The authors declare no competing interests.
