## [Peer Review File · Nature Communications]

LKRSDH-dependent histone modifications of insulin-like peptide sites contribute to age-related circadian rhythm changesReviewer #1 (Remarks to the Author):

Age-dependent changes in circadian rhythms have been observed in a variety of organisms including *Drosophila*, but the mechanisms that drive these changes during aging remain largely unknown. Here, the authors identify a relatively little studied bifunctional chromatin regulatory protein, LKRSDH (Lysine ketoglutarate reductase/saccharopine dehydrogenase), as a potential contributor to the age-dependent circadian behavior changes in flies. LKRSDH has been previously shown to inhibit H3K17me2 to inhibit gene expression, in addition to its enzymatic roles in lysine degradation. The authors use a rigorous genetic approach to show that LKRSDH is necessary for rhythmic behavior in old flies, and that its proper expression levels are important for this because overexpression also results in alterations in rhythmic behaviors. They use RNA-seq to identify genes with altered expression in LKRSDH mutants and show that several insulin-like peptide genes are upregulated, and that mutations in one of these genes, *ilp-5*, can rescue the change in rhythmic behavior resulting from loss of LKRSDH. While LKRSDH has been implicated in inhibition of H3K17me2 levels, the authors identified another histone repressive mark, H3K27me3, that normally increases upon aging but is reduced in old LKRSDH flies. They further examine genome-wide localization of H3K17me2 and H3K27me3 using ChIP-seq, providing some support for a model in which these two marks begin to function together in old flies. Overall – I was very excited by the author's data and I think this is potentially a very interesting set of observations. I think the text and figures would benefit from some editing to clarify and highlight several of these observations, and some of the RNA-seq and ChIP-seq analysis could be expanded so that it's a bit easier to follow. I'm also unsure about the model proposed by the authors, and I think it would be useful to consider if the enzymatic activity of LKRSDH could contribute to modulation of H3K27me3 levels via an alternative mechanism. Most of my comments (see below) could be addressed by text revisions and inclusion of some new analysis figures, and I do not think that new experiments would be necessary to address my concerns.

1. The authors identified LKRSDH in a screen, but they do not provide any details about the screen or how this was performed. I think this information would be helpful to place the work in context, and the authors should include a brief description of how many fly lines were screened and the approach used.
2. The nature of the LKRSDH allele used is not described initially in the text, and only a brief description is provided later in the results section. Could the authors clarify if this allele completely disrupts LKRSDH or is a partial loss of function. Were homozygous or heterozygous flies used for the experiments? In Table 1 – it looked to me like LKRSDH[MB01942] homozygous mutant flies were used, but in Table 2 - LKRSDH[MB01942] heterozygotes are shown with the *dilp5* mutant. Could the authors explain this clearly in the text and include a legend to describe Table 1 and 2 clearly. The language in the text sometimes refers to this allele as a knockdown rather than knockout – but I usually think of knockdown lines as referring to RNAi. Figure 4A qPCR data indicate that LKRSDH expression in the mutant line is ~50% of WT – suggesting this is a partial loss of function? Showing these qPCR data for LKRSDH earlier in the paper rather than with the *dilp* qPCR could be quite helpful.
3. The abstract needs to introduce the full name and function of LKRSDH – it is somewhat misleading to only describe one function of this protein, and the abbreviation won't be familiar to many people.
4. The genetics shown in Table 1, 2 and Figure 2 are complicated and will be quite difficult to follow for many readers. I appreciate the authors have been very careful to note the exact genotypes used – but I think the text needs to contain a little more detail about how this system works (line 110) – making it clear what the P{TOE.GS01508} line will do. If the figure labels for the graphs can be simplified -this would be very helpful.
5. Details for the RNA-seq and ChIP-seq studies need to be added to the methods including analysis approach used for RNA-seq, number of biological replicates, ZT used for collection times (this should also be mentioned in the text and figure legends). These data seem robust, but including those details is important.
6. Is LKRSDH itself significantly downregulated in the LKRSDH mutant vs *w1118* at either time point based on the RNA-seq? Showing LKRSDH on the RNA-seq log2 ratio plots (Fig3C/D) would be really helpful.
7. I suggest replacing "colocalized" (line 237) with "expressed in the same set of cells" or something similar.
8. It's pretty difficult to see a change in levels based on the ChIP-seq data shown in Figure 5A and

B. The correlation plots are also a bit difficult to read. Could the authors show these data as a heat map representing log₂FC over tss/gene body and/or as gene metaplots instead? The ChIP-qPCR data is much easier to follow (Fig 4L/M) and the western blots, particularly for H3K27me₃ (Fig 4C) are very convincing though.

9. In the discussion, the authors mention the limitation of the antibodies for ChIP-seq analysis, but in the results text (line 288) they suggest that "H3K17me₂ has a lower level of modification relative to H3K27me₃". The affinity and efficiency of the two antibodies are likely quite different, so I don't think it's appropriate to make conclusions about levels of these marks relative to each other. Comments regarding distribution are fine and supported by the data.

10. Based on the authors observations about the impact of LKRSDH on H3K27me₃, have they considered the possibility that this enzyme affects activity of the H3K27me₃ demethylase enzyme UTX? The JmjC demethylases require alpha-ketoglutarate for activity, so I am interested in whether LKRSDH's enzymatic activity could regulate H3K27me₃ levels via this mechanism. If appropriate, some discussion of this point could be included in the manuscript since the genetics approaches used to examine LKRSDH would affect both its functions.

Reviewer #2 (Remarks to the Author):

Comments for Author

Aging affects circadian rhythm in various organisms. However, the mechanisms have not been clearly revealed yet. The authors performed a screen to identify novel factors regulating the changes in circadian rhythm during aging. The manuscript presents that LKRSDH-dependent histone modification of insulin-like peptide genes contributes to age-related circadian rhythm changes. Their analysis for histone modification revealed the correlation between H3R17me₂ and H3K27me₃ in the aging process. Overall, the findings are interesting but leave some points unclear that temper my enthusiasm for the manuscript, as detailed below.

Major comments

1. The expression pattern of LKRSDH is analyzed in Figure 1, showing that it is expressed in neurons and glial cells. However, to show that LKRSDH mRNA or LKRSDH protein is actually expressed, in situ or immunostaining with antibody should be done. Also, would LKRSDH expression be observed in clock neurons or sleep-related neurons? Also, while Figure 2 shows that knockdown of LKRSDH in neurons results in a decrease in circadian rhythm, does knockdown of LKRSDH in clock neurons or sleep-related neurons also result in a similar decrease in circadian rhythm? More specific neurons for LKRSDH expression to regulate circadian rhythm need to be identified.

2. Figure 2B shows that knockdown of LKRSDH tends to suppress the circadian rhythm in aged individuals, but not significantly. It is possible that the knockdown efficiency of LKRSDH-RNAi is low. The knockdown efficiency needs to be shown. Also need to show if a similar trend is seen using different LKRSDH-RNAi lines.

3. In Figure S3B-D, the author concluded that the decreased lifespan of the LKRSDH mutant depends on the increased expression of dilp3 and dilp5. It is necessary to confirm whether the overexpression of dilp3 and/or dilp5 causes the decreased lifespan. Also, Figure S3D should use the LKDRSDH homozygous mutant, not the heterozygous flies.

4. Figure 2C shows that overexpression of LKRSDH has a negative effect on circadian rhythm as well as the knockdown. However, it is necessary to show experimentally how this can be explained in epigenetic terms.

5. In Figure 3, the authors focused on dilp3 and dilp5 from the transcriptional analysis. What are the other candidate genes than dilp3 or dilp5? The authors need to show the "narrow down" process of candidate factors.

6. It is interesting that the age-dependent decreased expression of dilp3 and dilp5 in Figure 4 can be explained by LKRSDH-dependent histone modification. How is the expression levels and histone modification of other dilps such as dilp2? Are they similar with dilp3/dilp5 regulation? In Figure 4A, aging decreases the expression in art4 as well as a decrease in LKRSDH; the reason why LKRSDH expression decreases with aging needs to be clarified. Furthermore, the authors need to confirm whether knockdown or mutants of LKRSDH and art4 show histone modification, decreased

expression of *dilp3/dilp5*, and altered circadian rhythm, as seen in aging.

7. Table 2 shows the genetic correlation between LKRSDH and *dilp3/dilp5* with respect to circadian rhythm. Since these are important data in this manuscript and should be shown in graphs like Figure 2 and Figure S1. Furthermore, does double mutant of *dilp3* and *dilp5* change circadian rhythm?

8. It is shown that change in H3R17me2 and H3K27me3 correlate with aging, however more detailed discussion is needed on what mechanism causes this.

Minor comments

9. Since sleep and longevity are greatly influenced by genetic background, the backgrounds of the comparisons must be the same. It should be noted how the backcrossing is done.

10. The authors should describe the RING assay method that is done in Figure S1J.

11. In Figure 2D, the UAS-*dCas9/P[TOE,GS01508]*; the *nSyb-Gal4/+* group has only one dot. If that is the case, the number of samples needs to be increased.

12. Figure 4B appears first in line 226, but Figure 4A should appear first.

13. The description of the LKRSDH mutant as knockdown should be corrected to mutant. (lines 257, 261, etc.)

Reviewer #3 (Remarks to the Author):

Lv and colleagues studied the role of LKRSDH, a histone H3R17me2 inhibitor, in mediating the impact of aging on circadian rhythm attenuation. They revealed that LKRSDH-dependent H3R17me2 and H3K27me3 modifications at the promoter region of insulin-like peptides *dilp3/5* is the underlying mechanism. The reviewer found interest in the findings, but throughout the manuscript, the conclusions have not been well supported. In addition, the statistical analyses were inappropriate.

Major concerns:

1. This paper consists of two parts; circadian behavioral rhythm alteration in LKRSDH mutant and epigenetic regulation of *dilp3/5* expression by LKRSDH. The findings from each part are interesting, but these lack a robust connection due to some pitfalls. For example, why is the effect of genetic manipulation of *dilp3/5* on circadian behavioral rhythms specific to aged flies? The authors raised the potential contribution of methyltransferase, but the hypothesis has not been tested.

2. Line 228: In relation above, the authors need to validate the hypothesis that decreased H3R17me2 levels in aged flies are dependent on the levels of arginine methyltransferase.

3. Statistical analysis: What is "One-way ANOVA with unpaired t-test"? The authors used this statistical analysis throughout the data analyses. The reviewer has no idea about the statistical analysis. Even worse, there is little information available about the statistical analysis in the methods. Next to impossible to understand how the authors performed the statistics on the data and believe the statistical significance throughout the data.

4. In Fig. S1, the authors found the decreased activity of circadian behavioral rhythms only in the young LKRSDH mutant but not the aged mutant. In contrast, Fig 2A showed reduced circadian behavioral rhythmicity in response to LKRSDH knockdown in neurons and glial cells only in the aged flies. Despite these inconsistent observations, the authors concluded that both are consistent phenotypes. This conclusion is not supported.

5. In relation to the above comment #3, the author needs to compare the efficiency of knockdown and overexpression of LKRSDH in neurons versus glial cells. This is a part of the reason that makes the conclusion unsupported and insignificant. In particular, the authors' conclusion "LKRSDH expressed by both neurons and glial cells is necessary for the maintenance of circadian rhythm in aged flies" is not supported well as the authors failed to see the recovery of circadian behavioral phenotypes by rescuing LKRSDH in neurons and glial cells. Overall, the findings from Fig 2 (and

supplemental figure 1) do not support each other, which reduces the scientific impact of the data.

6. Line 172: Based on the outcomes from DEG analysis, the authors concluded that LKRSDH serves as a transcriptional suppressor. This is a too strong conclusion and not supported by the data. There is a decent number of genes downregulated in the mutant compared with the wild type.

7. Line 294: How does LKRSDH knockdown or aging change genome-wide occupancy of H3R17me2 and H3K27me3? Fig. 5 and S5 did not show clear differences. Appropriate analyses or presentations would be required. Also, did ChIP-seq find statistical significance of the difference of the occupancy of H3R17me2 and H3K27me3 at *ilp3/5* promoters among each group of flies?

8. Figure 5C-F: The results are not significantly important. Both histone marks seem to be modified by some common catalytic enzymes, as the authors described. To stress the importance of the results, the authors also need to compare the correlation with other histone marks such as histone acetylation etc.

9. English needs to be significantly improved throughout the manuscript. For instance, it was so difficult or impossible for the reviewer to understand the messages from the authors in the paragraph starting at line 175.

10. Is the evidence consistent in mammals?

Minor concerns:

1. Fig. S1: Did the circadian behavioral rhythms change under LD? When did the authors measure the activity, at specific times of the day or over the circadian cycle (Fig. S1I)? What is the RING assay (Fig. S1J)? Is this a popular behavioral assay in flies? The authors need to describe the method so that non-drosophila researchers can understand the assay. Overall, figure legends and methods tended to lack essential information.

2. Fig. 2: How did the authors distinguish arrhythmic and rhythmic phenotypes? Any statistics? The reviewer would request to show representative behavioral data of each drosophila.

3. Fig. 4B and C: The authors should use H3 histone as a loading control, instead of tubulin.

4. As *dilp3/5* are involved in insulin-glucose metabolism, the authors should examine the circadian dynamics of insulin and glucose levels in each group of flies.

5. Information about the Pearson correlation analysis should move from the Result section to the Method section.

Reviewer #1 (Remarks to the Author):

Age-dependent changes in circadian rhythms have been observed in a variety of organisms including *Drosophila*, but the mechanisms that drive these changes during aging remain largely unknown. Here, the authors identify a relatively little studied bifunctional chromatin regulatory protein, LKRSDH (Lysine ketoglutarate reductase/saccharopine dehydrogenase), as a potential contributor to the age-dependent circadian behavior changes in flies. LKRSDH has been previously shown to inhibit H3K17me2 to inhibit gene expression, in addition to its enzymatic roles in lysine degradation. The authors use a rigorous genetic approach to show that LKRSDH is necessary for rhythmic behavior in old flies, and that its proper expression levels are important for this because overexpression also results in alterations in rhythmic behaviors. They use RNA-seq to identify genes with altered expression in LKRSDH mutants and show that several insulin-like peptide genes are upregulated, and that mutations in one of these genes, *ilp-5*, can rescue the change in rhythmic behavior resulting from loss of LKRSDH. While LKRSDH has been implicated in inhibition of H3K17me2 levels, the authors identified another histone repressive mark, H3K27me3, that normally increases upon aging but is reduced in old LKRSDH flies. They further examine genome-wide localization of H3K17me2 and H3K27me3 using ChIP-seq, providing some support for a model in which these two marks begin to function together in old flies. Overall – I was very excited by the author's data and I think this is potentially a very interesting set of observations. I think the text and figures would benefit from some editing to clarify and highlight several of these observations, and some of the RNA-seq and ChIP-seq analysis could be expanded so that it's a bit easier to follow. I'm also unsure about the model proposed by the authors, and I think it would be useful to consider if the enzymatic activity of LKRSDH could contribute to modulation of H3K27me3 levels via an alternative mechanism. Most of my comments (see below) could be addressed by text revisions and inclusion of some new analysis figures, and I do not think that new experiments would be necessary to address my concerns.

1. The authors identified LKRSDH in a screen, but they do not provide any details about the screen or how this was performed. I think this information would be helpful to place the work in context, and the authors should include a brief description of how many fly lines were screened and the approach used.

Answer: Thank you for bringing this to our attention. In the revised version, we mentioned in the first sentence of the last paragraph of the introduction section that we screened for mutants that affected age-dependent circadian rhythm changes from the *Drosophila* mutant library, which contain mutants for genes regulating histone methylation.

2. The nature of the LKRSDH allele used is not described initially in the text, and only a brief description is provided later in the results section. Could the authors clarify if this allele completely disrupts LKRSDH or is a partial loss of function. Were homozygous or heterozygous flies used for the experiments? In Table 1 – it looked to me like LKRSDH[MB01942] homozygous mutant flies were used, but in Table 2 - LKRSDH[MB01942] heterozygotes are shown with the *dilp5* mutant. Could the authors explain this clearly in the text and include a legend to describe Table 1 and 2 clearly. The language in the text

sometimes refers to this allele as a knockdown rather than knockout – but I usually think of knockdown lines as referring to RNAi. Figure 4A qPCR data indicate that LKRSDH expression in the mutant line is ~50% of WT – suggesting this is a partial loss of function? Showing these qPCR data for LKRSDH earlier in the paper rather than with the dilp qPCR could be quite helpful.

Answer: Thank you for the suggestion. We have added the RT-PCR results to verify the expression level of LKRSDH in *LKRSDH*^{MB01942} homozygous allele in figure S1I, along with corresponding descriptions. This allele exhibits a partial loss of function. LKRSDH[MB01942] is a mutant constructed by inserting Mi{ET1}, resulting in premature translation termination and abnormal LKRSDH function. In the second row of table 1 (and the corresponding row for 40d) and Figure S1, we utilized homozygous mutant. Heterozygous were used elsewhere as the genotypes indicated. We have clarified this in the revised manuscript. To be consistent, we now refer to the homozygous genotype as “*LKRSDH*^{MB01942} homozygous allele” in the descriptions, and the heterozygous genotype as “*LKRSDH*^{MB01942} mutant”.

3. The abstract needs to introduce the full name and function of LKRSDH – it is somewhat misleading to only describe one function of this protein, and the abbreviation won't be familiar to many people.

Answer: Thank you for the suggestion. We included the full name and a brief introduction of the function of LKRSDH in the abstract.

4. The genetics shown in Table 1, 2 and Figure 2 are complicated and will be quite difficult to follow for many readers. I appreciate the authors have been very careful to note the exact genotypes used – but I think the text needs to contain a little more detail about how this system works (line 110) – making it clear what the P{TOE.GS01508} line will do. If the figure labels for the graphs can be simplified -this would be very helpful.

Answer: Thank you for the suggestion. Descriptions of how P{TOE.GS01508} line work were added to the beginning of the second paragraph of the result session. We simplified the genetic labels as *LKRSDH* OE in the figure labels of the revised manuscript.

5. Details for the RNA-seq and ChIP-seq studies need to be added to the methods including analysis approach used for RNA-seq, number of biological replicates, ZT used for collection times (this should also be mentioned in the text and figure legends). These data seem robust, but including those details is important.

Answer: Thank you for the suggestion. The analysis approach used for RNA-seq, the number of biological replicates, and the ZT used for collection times have been added to method session under the title: RNA-seq, ChIP seq and data analysis. The ZT used for collection times has also been included in the corresponding text.

6. Is LKRSDH itself significantly downregulated in the LKRSDH mutant vs w1118 at either time point based on the RNA-seq? Showing LKRSDH on the RNA-seq log2 ratio plots (Fig3C/D) would be really helpful.

Answer: Thank you for the suggestion. We have found that the effective transcripts were significantly down regulated. This was confirmed through analysis of the RNAseq data. We calculated the ratio of the *LKRSDH* expression level in the mutant to the *w*¹¹¹⁸ which indicated the expression changes caused by the mutation. While the overall counts showed increase in the *LKRSDH* mutant compared to *w*¹¹¹⁸ control (Ratio>1), a significant downregulation was observed when analyzing the transcripts 5' to the insertion (Ratio<1), indicating a significant

decrease in the effective transcripts. The results were shown below. These findings have been included in Figure 3C, D in the revised version.

7. I suggest replacing “colocalized” (line 237) with “expressed in the same set of cells” or something similar.

Answer: Thank you for the suggestion. I agree with you. “colocalized” (line 237) has been replaced with “expressed in the same set of cells” in the revised manuscript.

8. It's pretty difficult to see a change in levels based on the ChIP-seq data shown in Figure 5A and B. The correlation plots are also a bit difficult to read. Could the authors show these data as a heat map representing log₂FC over tss/gene body and/or as gene metaplots instead? The ChIP-qPCR data is much easier to follow (Fig 4L/M) and the western blots, particularly for H3K27me3 (Fig 4C) are very convincing though.

Answer: Thank you for the suggestion. We changed Figure 5A, B into heat maps. Figure 5A is represented by figure 7A-D in the revised version. Figure 5B is represented by Figure 7E-G in the revised version. The correlation plots have also been improved. We added lines on the plot for better annotation.

9. In the discussion, the authors mention the limitation of the antibodies for ChIP-seq analysis, but in the results text (line 288) they suggest that “H3K17me2 has a lower level of modification relative to H3K27me3”. The affinity and efficiency of the two antibodies are likely quite different, so I don't think it's appropriate to make conclusions about levels of these marks relative to each other. Comments regarding distribution are fine and supported by the data.

Answer: Thank you for the suggestion. I agree. We revised the result session. The statement has been deleted.

10. Based on the authors observations about the impact of LKRSDH on H3K27me3, have they considered the possibility that this enzyme affects activity of the H3K27me3 demethylase enzyme UTX? The JmjC demethylases require alpha-ketoglutarate for activity, so I am interested in whether LKRSDH's enzymatic activity could regulate H3K27me3 levels via this mechanism. If appropriate, some discussion of this point could be included in the manuscript since the genetics approaches used to examine LKRSDH would affect both its functions.

Answer: I agree. Thank you for this insightful suggestion. We have incorporated a paragraph about this in the discussion session.

“Regarding the mechanism of LKRSDH's regulation of H3K27me3, our results suggested that this

regulation was dependent on *art4*. Evidence supporting this includes the observation that the presence or absence of *art4* changed the trend of H3K27me3 at the *ilps* locus in the background of the *LKRSDH* mutant (Figure S6Q-V), suggesting that *art4* was downstream of *LKRSDH*. In addition, it was found that the regulation of JH by *LKRSDH* was independent of its enzymatic activity²⁸. However, other mechanisms may also contribute. For instance, *UTX*, an alpha-ketoglutarate-dependent H3K27me3 demethylase enzyme, may be influenced by *LKRSDH*, thus affecting H3K27me3 levels. Further investigation could involve manipulating the enzyme-related domain to explore the involvement of *LKRSDH* in modulating *UTX*.”

Reviewer #2 (Remarks to the Author):

Comments for Author

Aging affects circadian rhythm in various organisms. However, the mechanisms have not been clearly revealed yet. The authors performed a screen to identify novel factors regulating the changes in circadian rhythm during aging. The manuscript presents that *LKRSDH*-dependent histone modification of insulin-like peptide genes contributes to age-related circadian rhythm changes. Their analysis for histone modification revealed the correlation between H3R17me2 and H3K27me3 in the aging process. Overall, the findings are interesting but leave some points unclear that temper my enthusiasm for the manuscript, as detailed below.

Major comments

1. The expression pattern of *LKRSDH* is analyzed in Figure 1, showing that it is expressed in neurons and glial cells. However, to show that *LKRSDH* mRNA or *LKRSDH* protein is actually expressed, in situ or immunostaining with antibody should be done. Also, would *LKRSDH* expression be observed in clock neurons or sleep-related neurons? Also, while Figure 2 shows that knockdown of *LKRSDH* in neurons results in a decrease in circadian rhythm, does knockdown of *LKRSDH* in clock neurons or sleep-related neurons also result in a similar decrease in circadian rhythm? More specific neurons for *LKRSDH* expression to regulate circadian rhythm need to be identified.

Answer: Thank you for the suggestion. We conducted an analysis of the expression pattern of endogenous *LKRSDH* using in situ. The results indicated that the *LKRSDH* exhibits widespread expression, which is well colocalized with the GFP signal in the *LKRSDH*-Gal4 constructed via the CRISPR/Cas-9 drop-in technique described in Kanca et al used in this study. We have included these in situ results in the updated Figure 1.

Furthermore, through the colocalization experiments involving *tim*-Gal4 and *LKRSDH*^{MB01942}, we observed their colocalization in a subset of neurons. Indeed, the knockdown of *LKRSDH* in clock neurons using *tim*-Gal4 results in comparable phenotype to the mutant allele. These results, along with other data from our study, indicated that in *LKRSDH* expressing neurons, the specific neurons for circadian rhythm regulation include clock neurons and the *ilp*-expressing neurons. We have provided visual representation of this information in supplementary Figure S8 and also discussed it extensively in the discussion session.

Kanca, O. et al. An efficient CRISPR-based strategy to insert small and large fragments of DNA

using short homology arms. *Elife* 8 (2019).

2. Figure 2B shows that knockdown of LKRSDH tends to suppress the circadian rhythm in aged individuals, but not significantly. It is possible that the knockdown efficiency of LKRSDH-RNAi is low. The knockdown efficiency needs to be shown. Also need to show if a similar trend is seen using different LKRSDH-RNAi lines.

Answer: Thank you for the suggestion. We have confirmed these results by using two additional LKRSDH RNAi lines. The knockdown efficiency has been verified (results shown in Figure S2K-M). Furthermore, similar phenotypic trends were observed using different LKRSDH-RNAi lines (results shown in Figure 2P-Q).

3. In Figure S3B-D, the author concluded that the decreased lifespan of the LKRSDH mutant depends on the increased expression of dilp3 and dilp5. It is necessary to confirm whether the overexpression of dilp3 and/or dilp5 causes the decreased lifespan. Also, Figure S3D should use the LKDRSDH homozygous mutant, not the heterozygous flies.

Answer: Thank you for the suggestion. In the revised version, we have examined the lifespan of flies overexpressing dilp3. The results have been included in Figure S4D.

We apologize for not being able to complete the experiment in Figure S3D with LKDRSDH homozygous mutants due to time constraints. We encountered difficulties in combining LKRSDH allele and dilp1-5^{KO} with balancers into a single fly strain. After trying with different balancers, we have just finished creating the required fly line and have started the final cross for the lifespan test. It will take an additional three months to complete. However, we do not anticipate a significant difference in the outcome. Because *LKDRSDH* allele is a hypomorph, demonstrating a decrease in the expression level for both homozygous and heterozygous alleles (Figure below). Based on the current information we have obtained, the homozygous and heterozygous mutants exhibit similar phenotype in terms of lifespan and circadian rhythm. Therefore, we speculate that using the homozygous mutant will yield similar results. We are in the process of conducting this experiment and would be happy to include the findings in the final version if necessary.

4. Figure 2C shows that overexpression of LKRSDH has a negative effect on circadian rhythm as well as the knockdown. However, it is necessary to show experimentally how this can be

explained in epigenetic terms.

Answer: Thank you for the suggestion. In response to your recommendation, we performed ChIP assay under the condition of LKRSDH overexpression. The results have been included in Figure S5E-J, along with corresponding descriptions.

5. In Figure 3, the authors focused on *dilp3* and *dilp5* from the transcriptional analysis. What are the other candidate genes than *dilp3* or *dilp5*? The authors need to show the “narrow down” process of candidate factors.

Answer: Thank you for the suggestion. Based on the enrichment analysis, we made the decision to study these genes. Notably, *dilp3* and *dilp5* were found to be significantly enriched in the longevity regulating pathway, which further supported their relevance and importance. During the revision process, we carefully examined the expression levels of all *dilp* family members and discovered that *dilp2*, *dilp3*, *dilp5* showed notable associations. To reflect these findings, we added the following sentences in the third part of the results section.

“Among the three gene profiles enriched in the longevity regulating pathway, we identified *dilp3* and *dilp5* on the list (Table S2). Then we examined the expression level of all the *dilp* family members and found that the *dilp2*, *dilp3*, *dilp5* exhibited changes in their expression levels under these conditions. Consequently, we conducted detailed studies on the relationship between *LKRSDH* and these *dilp* genes.”

6. It is interesting that the age-dependent decreased expression of *dilp3* and *dilp5* in Figure 4 can be explained by LKRSDH-dependent histone modification. How is the expression levels and histone modification of other *dilps* such as *dilp2*? Are they similar with *dilp3/dilp5* regulation? In Figure 4A, aging decreases the expression in *art4* as well as a decrease in LKRSDH; the reason why LKRSDH expression decreases with aging needs to be clarified. Furthermore, the authors need to confirm whether knockdown or mutants of LKRSDH and *art4* show histone modification, decreased expression of *dilp3/dilp5*, and altered circadian rhythm, as seen in aging.

Answer: Thank you for the suggestion.

First, concerning the other *dilps*, the *dilp2* was found to be up regulated in aged flies, while *dilp1* and *dilp4* were unaffected under the conditions we examined. This result has been included in figure 4A and Figure S3B. The data of histone modifications at these sites have been presented in Figure 6K-M and Figure S7D-E. We appreciate the suggestion, as it led us to discover that the *dilp2* shows a significant increase in aged flies. Based on these finding, we propose that the significant changes in total *dilp* levels in aged flies, caused by *LKRSDH* mutants, accounted for its age specific circadian phenotypes.

Secondly, why dose LKRSDH expression decrease with aging? This could be partially explained by down regulation of *art4* during aging. Real time RT-PCR showed that knocking down of *art4* results in down regulation of *LKRSDH* expression (Figure S6W in the revised manuscript), suggesting that a feedback mechanism exist at transcriptional level. It seems that the machinery related to H3R17me2 were down regulated in aged flies, as the levels of modification, *art4*, *LKRSDH* were all down regulated. This has been included in the revised version.

Thirdly, down regulation of *art4* results in decreased H3R17me2, increased H3K27me3, down regulation of expression of *dilp3/dilp5*. This result has been put into Figure S6. In respond to

your suggestion, we have examined the circadian phenotypes caused by the up or down regulating of *art4* (Figure 5). Genetic interaction experiments indicated that the down regulation of *art4* results in the suppression of phenotypes caused by *LKRSDH* mutation (Figure 5C-D).

7. Table 2 shows the genetic correlation between *LKRSDH* and *dilp3/dilp5* with respect to circadian rhythm. Since these are important data in this manuscript and should be shown in graphs like Figure 2 and Figure S1. Furthermore, does double mutant of *dilp3* and *dilp5* change circadian rhythm?

Answer: Thank you for your suggestion. The Table 2 has been changed to Figure 4 in the revised version. As recommended by the reviewer, we examined the expression levels of all *dilp* family members and found that *dilp2, 3, 5* were relevant. Therefore, we investigated the genetic interaction between *LKRSDH* and *dilp2, 3, 5*. The circadian phenotype of double mutant of *dilp3* and *dilp5* was also shown in Figure 4.

8. It is shown that change in H3R17me2 and H3K27me3 correlate with aging, however more detailed discussion is needed on what mechanism causes this.

Answer: Thank you for your suggestion. The following discussions has been added.

“Our results indicated that changes in H3R17me2 and H3K27me3 correlate with aging. Previous studies have shown that histone modifications, such as H4K16 acetylation, H3K4 trimethylation, and H3K9 or H3K27 trimethylation, serve as hallmarks of aging. These modifications can lead to transcriptional changes, loss of cellular homeostasis, and age-associated metabolic decline. These findings suggest that H3R17me2 may have similar effects on the aging process as other histone modifications and may also serve as a hallmark of aging. Our study further revealed that H3R17me2 specifically correlates with aging. The regulation of aging-related H3R17me2 changes is influenced by *LKRSDH*. Lifespan was found to be affected by *LKRSDH* mutation. Previous reports have indicated that *art4*, which is responsible for H3R17me2, is linked to the aging process. These evidences suggested that the mechanism for H3R17me2 and H3K27me3 changes during aging is linked with the writer or modifiers of them, which may be influenced by the changes of metabolic or signaling pathways outside of the nucleus during aging process.”

Ref: Oh, E.S. & Petronis, A. Origins of human disease: the chronoepigenetic perspective. *Nat. Rev. Genet.* **22**, 533–546 (2021).

Minor comments

9. Since sleep and longevity are greatly influenced by genetic background, the backgrounds of the comparisons must be the same. It should be noted how the backcrossing is done.

Answer: Thank you for pointing out this. This has been added to the end of the first paragraph of the methods session.

10. The authors should describe the RING assay method that is done in FigureS1J.

Answer: Thank you for pointing out this. Session “Rapid Iterative Negative Geotaxis (RING Assay)” has been made to the methods.

11. In Figure 2D, the UAS-dCas9/P[TOE,GS01508]; the nSyb-Gal4/+ group has only one dot. If that is the case, the number of samples needs to be increased.

Answer: Thank you for pointing out this. The problem here was that the rhythmicity was very low for this genotype. Since this figure only counts the flies with normal rhythmicity, the number shown here was low. We repeated the experiment with more flies and increased the number of flies that can be shown here.

12. Figure 4B appears first in line 226, but Figure 4A should appear first.

Answer: Thank you for pointing out this. Figure 4A appeared in the first paragraph in the result section with the title: “*Dilp3* and *dilp5* contribute to the circadian rhythm and longevity phenotypes caused by *LKRSDH* mutation.”

13. The description of the LKRSDH mutant as knockdown should be corrected to mutant. (lines 257, 261, etc.)

Answer: Thank you for pointing out this. This has been corrected in the revised version.

Reviewer #3 (Remarks to the Author):

Lv and colleagues studied the role of LKRSDH, a histone H3R17me2 inhibitor, in mediating the impact of aging on circadian rhythm attenuation. They revealed that LKRSDH-dependent H3R17me2 and H3K27me3 modifications at the promoter region of insulin-like peptides *dilp3/5* is the underlying mechanism. The reviewer found interest in the findings, but throughout the manuscript, the conclusions have not been well supported. In addition, the statistical analyses were inappropriate.

Major concerns:

1. This paper consists of two parts; circadian behavioral rhythm alteration in LKRSDH mutant and epigenetic regulation of *dilp3/5* expression by LKRSDH. The findings from each part are interesting, but these lack a robust connection due to some pitfalls. For example, why is the effect of genetic manipulation of *dilp3/5* on circadian behavioral rhythms specific to aged flies? The authors raised the potential contribution of methyltransferase, but the hypothesis has not been tested.

Answer: Thank you for pointing out this. We appreciate the suggestion. We made efforts to clarify the link between these two parts of the study by conducting additional experiments. Figure 5 was added to make a connection in the revised manuscript. The rationale here is that the significant changes in total insulin-like peptide (*dilp*) levels in aged flies, caused by *LKRSDH* mutants, account for their age specific circadian phenotypes. *LKRSDH* mutants caused changes in total *dilp* levels by affecting *art4* and *art4* dependent histone modifications, primarily H3K27me3 and H3R17me2.

As recommended by the reviewers, we tested the expression levels of all members of the *dilp* gene family. The results showed that *dilp2* exhibited a significant increase, especially in aged *LKRSDH* mutants. Along with the changes in *dilp3* and *dilp5*, these evidences can potentially explain why *LKRSDH* mutation only led to circadian rhythm defects in aged flies.

The genetic interaction experiment showed that *dilp2* mutant can rescue the *LKRSDH* mutant phenotype. In conclusion, the circadian rhythm phenotype specific to aged flies was caused by an age-specific high expression of *dilp*. Indeed, genetic manipulation of *dilp3/5* affected circadian behavioral rhythms specific to aged flies. This may be because aged flies had very low *dilps* levels. Same levels of overexpression results in a more significant increase relative to the endogenous levels in aged flies compared to the young flies.

As reported previously, *LKRSDH* acted as a transcriptional corepressor in ecdysone signaling. It inhibited CARMER (*art4* ortholog) binding on the chromosome, which resulted in a decrease in methylated histone H3 on R17. Therefore, we propose that *art4* is the link between *LKRSDH* and *dilps*. We did extra experiments to verify this. Several lines of evidences supported this idea. First, knocking down *art4* could partially rescue the phenotype caused by *LKRSDH* mutation in the aged flies (Figure 5C-D). Second, overexpression of *art4* results in a significant down regulation of rhythmicity, especially in aged flies (Figure 5A-B). Third, the *dilps* showed consistent changes in the down regulation or upregulation of *art4*, respectively (Figure 5E, F). Fourth, the global level of H3R17me2 and H3K27me3 and the levels of them at the *dilp* locus can be regulated by either *LKRSDH* (Figure 6C, K-6M) or *art4* (Figure S6). These evidences were added to the revised manuscript.

2. Line 228: In relation above, the authors need to validate the hypothesis that decreased H3R17me2 levels in aged flies are dependent on the levels of arginine methyltransferase.

Answer: Thank you for the suggestion. To validate that the levels of H3R17me2 in aged flies are dependent on the levels of arginine methyltransferase, we conducted additional experiments. The evidences indicated that down regulation of *art4* by RNAi leads to a decrease in H3R17me2 both genome-wide (Figure S6C) and at *ilp* locus (Figure S6E-G). Consistently, the overexpression of *art4* led to an increase in H3R17me2 both genome wide (Figure S6D) and at *ilp* locus (Figure S6K-P).

3. Statistical analysis: What is "One-way ANOVA with unpaired t-test"? The authors used this statistical analysis throughout the data analyses. The reviewer has no idea about the statistical analysis. Even worse, there is little information available about the statistical analysis in the methods. Next to impossible to understand how the authors performed the statistics on the data and believe the statistical significance throughout the data.

Answer: Thank you for pointing out this. We apologize for the mistake in the text. The mistake has been rectified in the updated version. We have also included information about the statistical analysis in each section of the methods session.

4. In Fig. S1, the authors found the decreased activity of circadian behavioral rhythms only in the young *LKRSDH* mutant but not the aged mutant. In contrast, Fig 2A showed reduced circadian behavioral rhythmicity in response to *LKRSDH* knockdown in neurons and glial cells only in the aged flies. Despite these inconsistent observations, the authors concluded that both are consistent phenotypes. This conclusion is not supported.

Answer: Although the circadian rhythm in *Drosophila* is typically measured by examining the locomotion activity, it should be noted that the activity does not always exhibit the same trend of changes as the circadian rhythm. This phenomenon has also been demonstrated in other studies (References can be found below). The main reason for this discrepancy lies in the fact

that the Activity parameter here reflects the overall activity level, while the parameters in circadian rhythm primarily reflect the circadian rhythm aspects of the activity.

Klarsfeld, A., Leloup, J. C. & Rouyer, F. Circadian rhythms of locomotor activity in *Drosophila*. *Behavioural Processes* **64**, 161-175 (2003).

Maurer, C., Winter, T., Chen, S. W., Hung, H. C. & Weber, F. The CREB-binding protein affects the circadian regulation of behaviour. *Febs Letters* **590**, 3213-3220 (2016).

Rakshit, K., Wambua, R., Giebultowicz, T. M. & Giebultowicz, J. M. Effects of exercise on circadian rhythms and mobility in aging *Drosophila melanogaster*. *Experimental Gerontology* **48**, 1260-1265 (2013).

5. In relation to the above comment #3, the author needs to compare the efficiency of knockdown and overexpression of LKRSDH in neurons versus glial cells. This is a part of the reason that makes the conclusion unsupported and insignificant. In particular, the authors' conclusion "LKRSDH expressed by both neurons and glial cells is necessary for the maintenance of circadian rhythm in aged flies" is not supported well as the authors failed to see the recovery of circadian behavioral phenotypes by rescuing LKRSDH in neurons and glial cells. Overall, the findings from Fig 2 (and supplemental figure 1) do not support each other, which reduces the scientific impact of the data.

Answer: Thank you for the suggestion. We apologize for any unclear descriptions in this part. The *LKRSDH* mutant phenotype can be rescued by the re expression of *LKRSDH* by *LKRSDH*-Gal4. This set of experiments aims to determine the specific cells in which *LKRSDH* function. Our observations revealed the presence of both neurons and glial cells expressing *LKRSDH*. To investigate further, we utilized repo-Gal4 and nSyb-Gal4 to manipulate the expression levels of *LKRSDH* and assess their respective significance. RNAi of *LKRSDH* recapitulates a part of the mutant phenotype, that's why we conclude that LKRSDH expressed by both neurons and glial cells is necessary for the maintenance of circadian rhythm in aged flies.

However, the overexpressing *LKRSDH* either in all neurons or glia results in new functions. Moreover, neither the exclusive expression of *LKRSDH* in neurons nor in glial cells alone could rescue the phenotype. This suggests that the presence of *LKRSDH* in both neurons and glial cells is indispensable for the maintenance of circadian rhythm in aged flies. We have made this clear in the revised manuscript. The efficiency of knockdown and overexpression of *LKRSDH* in neurons versus glial cells is compared below. The other reviewer recommended to used multiple RNAi lines. These new sets of experiments were also added to the revised manuscript.

6. Line 172: Based on the outcomes from DEG analysis, the authors concluded that LKRSDH serves as a transcriptional suppressor. This is a too strong conclusion and not supported by the data. There is a decent number of genes downregulated in the mutant compared with the wild type.

Answer: Yes. I agree with you. Thank you for the suggestion. This has been revised.

7. Line 294: How does LKRSDH knockdown or aging change genome-wide occupancy of H3R17me2 and H3K27me3? Fig. 5 and S5 did not show clear differences. Appropriate analyses or presentations would be required. Also, did ChIP-seq find statistical significance of the difference of the occupancy of H3KR17me2 and H3K27me3 at ilp3/5 promoters among each group of flies?

Answer: Thank you for the suggestion. The other reviewer also suggested changing the format of Figure 5A to better represent the data. We have made changes to the format of Figure 5A, B, and S5A (now Figure 7A-D in the revised version) in the updated manuscript. Additionally, we have included the occupancy of H3KR17me2 and H3K27me3 at dilp2/3/5 promoters, as obtained from the CHIP-seq data, in Figure 7E, F, G of the revised manuscript.

8. Figure 5C-F: The results are not significantly important. Both histone marks seem to be modified by some common catalytic enzymes, as the authors described. To stress the importance of the results, the authors also need to compare the correlation with other histone marks such as histone acetylation etc.

Answer: Thank you for the suggestion. It's a good point to compare the correlation with other histone marks to emphasize the significance of the results. In the revised version, we have added a correlation analysis with other histone marks.

The catalytic enzyme for H3R17me2 is *art4* (Cakouros, D. et al., 2004), while the catalytic enzyme for H3K27me3 is E(Z) (Czermin, B. et al., 2002). The cross talk between these two markers is dependent on LKRSDH and *art4* (Cakouros, D. et al., 2008; Karakashev, S et al., 2018). Consistently, we found that LKRSDH which affected *art4* binding on chromosome affected both these histone modifications (Figure 6). That's why we focused on these two modifications in ChIP-seq experiments. We apologize for our resource limitations, which prevented us from conducting ChIP-seq for all other histone modifications. However, we obtained information on other histone modifications from existing datasets available in NCBI and performed some analysis.

As suggested, we checked the correlation of these two modifications with other modifications such as H3K27ac, H3K4me3, H3K9me3 and H3K36me3 obtained from *Drosophila* neurons directing reproductive behavior in a previous publication (GSE153901, Palmateer CM et al., 2021; GSE37032, Henry GL et al., 2012; GSE59769, Schertel C et al., 2015; GSE94922, Pascual-Garcia, P et al., 2017). The results of this analysis are shown in Figure S7E-T.

References

- Czermin, B., Melfi, R., McCabe, D., Seitz, V., Imhof, A., Pirrotta, V. (2002). *Drosophila* enhancer of Zeste/ESC complexes have a histone H3 methyltransferase activity that marks chromosomal Polycomb sites. *Cell* 111(2): 185--196.
- Henry, G. L., Davis, F. P., Picard, S., & Eddy, S. R. (2012). Cell type-specific genomics of *Drosophila* neurons. *Nucleic acids research*, 40(19), 9691–9704.
- Schertel, C., Albarca, M., Rockel-Bauer, C., Kelley, N. W., Bischof, J., Hens, K., van Nimwegen, E., Basler, K., & Deplancke, B. (2015). A large-scale, in vivo transcription factor screen defines bivalent chromatin as a key property of regulatory factors mediating *Drosophila* wing development. *Genome research*, 25(4), 514–523.
- Pascual-Garcia, P., Debo, B., Aleman, J. R., Talamas, J. A., Lan, Y., Nguyen, N. H., Won, K. J., & Capelson, M. (2017). Metazoan Nuclear Pores Provide a Scaffold for Poised Genes and Mediate Induced Enhancer-Promoter Contacts. *Molecular cell*, 66(1), 63–76.e6. <https://doi.org/10.1016/j.molcel.2017.02.020>
- Cakouros, D., Daish, T. J., Mills, K. & Kumar, S. An arginine-histone methyltransferase, 626 CARMER, coordinates ecdysone-mediated apoptosis in *Drosophila* cells. *J. Biol. Chem.* **279**, 627 18467–18471 (2004).
- Cakouros, D. et al. dLKR/SDH regulates hormone-mediated histone arginine methylation and transcription of cell death genes. *J. Cell. Biol.* **182**, 481–495 (2008).
- Karakashev, S. et al. CARM1-expressing ovarian cancer depends on the histone methyltransferase EZH2 activity. *Nat. Commun.* **9**, 631 (2018).
- Palmateer CM, Moseley SC, Ray S, Brovero SG et al. Analysis of cell-type-specific chromatin modifications and gene expression in *Drosophila* neurons that direct reproductive behavior. *PLoS Genet* 2021 Apr;17(4):e1009240.

9. English needs to be significantly improved throughout the manuscript. For instance, it was so difficult or impossible for the reviewer to understand the messages from the authors in the paragraph starting at line 175.

Answer: We apologize for this. We have made significant efforts to improve the language of the manuscript, including the paragraph starting at line 175. The manuscript has undergone a comprehensive revision. We apologize if the language may not as accurate as that of a native English speaker.

10. Is the evidence consistent in mammals?

Answer: It's an intriguing problem that requires further investigation. It's beyond our ability to solve it in this study. It's not clear if the mechanism involving LKRSDH-art4-ilps found in this study were conserved in mammals. However, we have some clues pointing to consistent evidence of the mechanisms found in this study in mammals. We added evidences in the paragraph of discussion about this problem in the revised manuscript.

"Our results are consistent with previous studies investigating the impact of *insulin* on aging. *LKRSDH* mutation resulted in increased *dilps* and reduced lifespan. Previous research has demonstrated that decreased insulin pathway activity results in a longer lifespan. A study in pancreatic cells indicated that H3R17me2 was involved in the regulation of insulin. Therefore, this study revealed the upstream regulation of *dilps*, which may be conserved and influential in vertebrates."

Minor concerns:

1. Fig. S1: Did the circadian behavioral rhythms change under LD? When did the authors measure the activity, at specific times of the day or over the circadian cycle (Fig. S1I)? What is the RING assay (Fig. S1J)? Is this a popular behavioral assay in flies? The authors need to describe the method so that non-drosophila researchers can understand the assay. Overall, figure legends and methods tended to lack essential information.

Answer: Thank you for your suggestion. We have examined the changes in circadian rhythms under LD conditions. The results have been presented in figure S1L-M. The activity data was obtained from over the circadian cycle. We have provided a more comprehensive description of the RING assay in the corresponding methods and figure legends.

2. Fig. 2: How did the authors distinguish arrhythmic and rhythmic phenotypes? Any statistics? The reviewer would request to show representative behavioral data of each drosophila.

Answer: Individual flies with power (≥ 10) and "width" value of 1.5 or more (representing the peak number within 30 minutes increments above the 95% confidence line of the periodogram) were considered rhythmic. The percentage of rhythmic and arrhythmic flies were calculated using a standard protocol widely used in the field. The original reference for this protocol has been previously published (References shown below). Further details regarding the method can be found in the Methods section. The original data of all individual flies were included in dataset 1.

Klarsfeld, A., Leloup, J. C. & Rouyer, F. Circadian rhythms of locomotor activity in *Drosophila*. *Behavioural Processes* **64**, 161-175 (2003).

3. Fig. 4B and C: The authors should use H3 histone as a loading control, instead of tubulin.

Answer: Thank you for the suggestion. We have repeated all the western blots with H3 histone as loading control.

4. As *dilp3/5* are involved in insulin-glucose metabolism, the authors should examine the circadian dynamics of insulin and glucose levels in each group of flies.

Answer: Thank you for the suggestion. We have examined the circadian dynamics of insulin and glucose levels in 3d *w¹¹¹⁸*, 3d *LKRSDH^{MB01942}*, 40d *w¹¹¹⁸* 和 40d *LKRSDH^{MB01942}*. The results have been included in Figure S3C-F of the revised manuscript.

5. Information about the Pearson correlation analysis should move from the Result section to the Method section.

Answer: Thank you for the suggestion. This has been revised in the updated version.

Reviewer #1 (Remarks to the Author):

The authors have addressed many of my original concerns, but the data presented in response to question #6 raises a very substantial concern for me. The RNA-seq data for LKRSDH in the LRKSDH (heterozygote?) vs w1118 does not show a significant decrease in levels, and I am not convinced by the graph showing a decrease only in the 5' region. Were the qPCR primers used to assess LKRSDH levels also in this 5' region? I can't see a significant difference in LKRSDH levels between these genotypes, and in the absence of data about protein levels - I find it difficult to attribute the phenotypes shown to decreased LKRSDH levels.

A more minor point is that the authors did not respond appropriately to my question #1 about the details for the screen. The sentence provided in the introduction does not give me any information about how the screen was conducted, how many lines were analyzed. I also don't quite understand why the authors switched from using the homozygous mutant they identified in the screen to using the heterozygote in all subsequent analysis - this was unclear in the first version due to the nomenclature used, and I don't understand the rationale behind this given that the phenotypes were identified first in the homozygous mutant.

Reviewer #2 (Remarks to the Author):

The revised version of the manuscript is significantly improved, and the authors have performed good experiments addressing most of my major concerns. Only one issue remains. The authors added the FISH data to show the expression pattern of LKRSDH. However, the expression pattern of LKRSDH shown by FISH is not similar with that of LKRSDH-Gal4. The authors need to explain the reason for such a difference anywhere.

Reviewer #3 (Remarks to the Author):

The authors have successfully addressed nearly all the concerns raised by the reviewer. Nevertheless, further improvements are necessary to elevate the language to an acceptable standard. Notably, there are numerous grammatical errors involving plural vs singular forms (line 491, these evidence's) and present vs past tenses (particularly evident in the second paragraph of the introduction).

It is highly recommended that the manuscript undergo a thorough review and revision by a native English writer or a large language model such as Chat-GPT before publication.

REVIEWER COMMENTS

Reviewer #1 (Remarks to the Author):

The authors have addressed many of my original concerns, but the data presented in response to question #6 raises a very substantial concern for me. The RNA-seq data for LKRSDH in the LKRSDH (heterozygote?) vs w1118 does not show a significant decrease in levels, and I am not convinced by the graph showing a decrease only in the 5' region. Were the qPCR primers used to assess LKRSDH levels also in this 5' region? I can't see a significant difference in LKRSDH levels between these genotypes, and in the absence of data about protein levels - I find it difficult to attribute the phenotypes shown to decreased LKRSDH levels.

A more minor point is that the authors did not respond appropriately to my question #1 about the details for the screen. The sentence provided in the introduction does not give me any information about how the screen was conducted, how many lines were analyzed. I also don't quite understand why the authors switched from using the homozygous mutant they identified in the screen to using the heterozygote in all subsequent analysis - this was unclear in the first version due to the nomenclature used, and I don't understand the rationale behind this given that the phenotypes were identified first in the homozygous mutant.

Answer: We sincerely appreciate your valuable suggestions. In this study, the qPCR primers were specifically designed to target the second and third exons of *LKRSDH*, which are situated in the 5' region of the gene. Notably, the corresponding qPCR analysis revealed a significant reduction in *LKRSDH* mRNA levels within the 5' region of the *LKRSDH*^{MB01942} mutant, strongly suggesting that the *LKRSDH*^{MB01942} mutation was caused by a large insertion (Figure S11). Additionally, to address your concern, we conducted a search for commercially available antibodies targeting the homologs of LKRSDH. We discovered an antibody against AASS (aminoadipate-semialdehyde synthase), which is the human equivalent of LKRSDH. This antibody was raised against a region that shares a certain level of identity (Please refer to the figure below) with the *Drosophila* LKRSDH protein. We performed experiments using the

antibody on *Drosophila* lysate and confirmed its ability to specifically detect the LKRSDH protein. In lysates from S2 cells overexpressing LKRSDH, the antibody successfully detected the tagged protein. Additionally, we observed a significant decrease in protein levels in the mutant samples. The quantification of LKRSDH protein levels in homozygous mutants, heterozygous mutants, and control samples has been included in Figure S1 along with corresponding descriptions.

We have incorporated more detailed information about the screen into the last paragraph of the introduction section. We created a list of genes associated with histone methylation based on a previous publication (Supplementary file 2 in Jenkins et al.¹). Mutants for each of these genes were individually selected and ordered from the Bloomington stock center. After obtaining the mutants, we first backcrossed them with *w¹¹¹⁸* to clean up the genetic background. We performed circadian rhythm assessments on 3-day-old flies, including both *w¹¹¹⁸* control flies and mutants. Simultaneously, we collected both *w¹¹¹⁸* control flies and mutants, which were subsequently subjected to a standardized aging program lasting for 40 days. We then evaluated the circadian rhythm of these flies. Compared with the control *w¹¹¹⁸*, we found that the circadian rhythm of *LKRSDH^{MB01942}* decreased significantly after aging.

The decision not to use the homozygous LKRSDH allele in genetic interaction experiments was primarily based on technical considerations and the availability of suitable genetic crosses. In order to perform genetic interaction experiments, it is necessary to combine multiple alleles or transgenes together on the same chromosome or fly.

In the case of Figure 4F, we initially used heterozygous mutants for convenience in genetic crosses because we encountered difficulties in obtaining lines with both the *LKRSDH* and *dilps* alleles together. The aim was to finally combine the homozygous alleles with other relevant alleles. Moreover, for Figure 1C, D, all four alleles/transgenes were already present on the same chromosome, making it technically impossible to include the homozygous LKRSDH allele in those specific experiments. However, we believe that these experiments with heterozygous were sufficient to for the conclusions we made in this study. In the phenotypes presented in Figure 1A and Figure 4D, as well as the Western blot (WB) data included in this version in Figure S1, both the homozygous and heterozygous mutants exhibited hypomorphic characteristics and showed comparable phenotypes.

Identity: 37.70%

Homo sapiens	DKKVLVLGAGMVSAPLVEWLHREKDVSI TVCSQVKEEADR	40
Drosophila melanogaster	RRKVLVLGSGYISEPVLEYSRDNIEITVGS DMKNQIEQ	40
Consensus	kvlvlg g s p e l r itv s k	
Homo sapiens	LAQQYAGVDSVYLDVNESTGHLQELCGRADV VVSLLPYSL	80
Drosophila melanogaster	LGKKYN. INPVSMDICKQEEKLGF LVAKQDLVISLLPYVL	79
Consensus	l y v d l l d v sllpy l	
Homo sapiens	HGMVARYCVAEGTHMVTASYLNDEISGLH EEAkakgvtim	120
Drosophila melanogaster	HPLVAKACITNKVN MVTASYITPALKELEKSV EDAGITII	119
Consensus	h va c mvtasy l g ti	
Homo sapiens	NE	122
Drosophila melanogaster	GE	121
Consensus	e	

Reference

1. Jenkins, A. M. & Muskavitch, M. A. T. Evolution of an Epigenetic Gene Ensemble within the Genus *Anopheles*. *Genome Biol. Evol.* **7**, 901-915 (2015).

Reviewer #2 (Remarks to the Author):

The revised version of the manuscript is significantly improved, and the authors have performed good experiments addressing most of my major concerns. Only one issue remains. The authors added the FISH data to show the expression pattern of LKRSDH. However, the expression pattern of LKRSDH shown by FISH is not similar with that of LKRSDH-Gal4. The authors need to explain the reason for such a difference anywhere.

Answer: Thank you for the suggestion. The Gal4 lines were generated by inserting a Gal4-expressing cassette into the genomic region of a specific gene. This approach captures a portion of the regulatory elements responsible for Gal4 expression. However, it is important to note that this method may not capture all the transcriptional regulation of the gene. Therefore, it is common that these Gal4 lines only represent a subset of the gene's overall expression. This observation has also been validated in previous publications (Figure 2F-H in Zhao et al.¹). This has been incorporated into the first paragraph under the title of "The expression of *LKRSDH* in both neurons and glial cells is necessary for maintaining circadian rhythm in aged flies" of the results session.

Reference

1. Zhao, Z. W. et al. Epigenetic regulator Stuxnet modulates octopamine effect on sleep through a Stuxnet-Polycomb-Oct β 2R cascade. *Embo Rep.* **22**, e47910 (2021).

Reviewer #3 (Remarks to the Author):

The authors have successfully addressed nearly all the concerns raised by the reviewer. Nevertheless, further improvements are necessary to elevate the language to an acceptable standard. Notably, there are numerous grammatical errors involving plural vs singular forms (line 491, these evidence's') and present vs past tenses (particularly evident in the second paragraph of the introduction).

It is highly recommended that the manuscript undergo a thorough review and revision by a native English writer or a large language model such as Chat-GPT before publication.

Answer: We sincerely appreciate your valuable suggestion. As per your recommendation, we have thoroughly revised the language of the manuscript using Chat-GPT, with a particular focus on ensuring correct usage of plural and singular forms, as well as distinctions between present and past tenses.

Reviewer #1 (Remarks to the Author):

The addition of the western blots to demonstrate decreased protein levels in the heterozygote and homozygote mutants has addressed my previous concerns about these mutants. The text explaining the mutant screen also helps address those concerns. I have no further concerns.

Reviewer #2 (Remarks to the Author):

The authors adequately responded to all my queries. I have no more comments on the revised manuscript.

REVIEWERS' COMMENTS

Reviewer #1 (Remarks to the Author):

The addition of the western blots to demonstrate decreased protein levels in the heterozygote and homozygote mutants has addressed my previous concerns about these mutants. The text explaining the mutant screen also helps address those concerns. I have no further concerns.

Response: We are delighted to have comprehensively addressed the concerns raised by the reviewer. Additionally, we express our sincere appreciation for the reviewer's insightful comments, which have significantly contributed to the reinforcement of our conclusions.

Reviewer #2 (Remarks to the Author):

The authors adequately responded to all my queries. I have no more comments on the revised manuscript.

Response: We appreciate the positive feedback from the reviewer on our revised manuscript.